# An Adaptive Empirical Bayesian Method for Sparse Deep Learning

**Wei Deng**
Department of Mathematics
Purdue University
West Lafayette, IN 47907
`deng106@purdue.edu`

**Xiao Zhang**
Department of Computer Science
Purdue University
West Lafayette, IN 47907
`zhang923@purdue.edu`

**Faming Liang**
Department of Statistics
Purdue University
West Lafayette, IN 47907
`fmliang@purdue.edu`

**Guang Lin**
Departments of Mathematics, Statistics
and School of Mechanical Engineering
Purdue University
West Lafayette, IN 47907
`guanglin@purdue.edu`

## Abstract

We propose a novel adaptive empirical Bayesian (AEB) method for sparse deep learning, where the sparsity is ensured via a class of self-adaptive spike-and-slab priors. The proposed method works by alternatively sampling from an adaptive hierarchical posterior distribution using stochastic gradient Markov Chain Monte Carlo (MCMC) and smoothly optimizing the hyperparameters using stochastic approximation (SA). We further prove the convergence of the proposed method to the asymptotically correct distribution under mild conditions. Empirical applications of the proposed method lead to the state-of-the-art performance on MNIST and Fashion MNIST with shallow convolutional neural networks (CNN) and the state-of-the-art compression performance on CIFAR10 with Residual Networks. The proposed method also improves resistance to adversarial attacks.

## 1 Introduction

MCMC, known for its asymptotic properties, has not been fully investigated in deep neural networks (DNNs) due to its unscalability in dealing with big data. Stochastic gradient Langevin dynamics (SGLD) [Welling and Teh, 2011], the first stochastic gradient MCMC (SG-MCMC) algorithm, tackled this issue by adding noise to the stochastic gradient, smoothing the transition between optimization and sampling and making MCMC scalable. Chen et al. [2014] proposed using stochastic gradient Hamiltonian Monte Carlo (SGHMC), the second-order SG-MCMC, which was shown to converge faster. In addition to modeling uncertainty, SG-MCMC also has remarkable non-convex optimization abilities. Raginsky et al. [2017], Xu et al. [2018] proved that SGLD, the first-order SG-MCMC, is guaranteed to converge to an approximate global minimum of the empirical risk in finite time. Zhang et al. [2017] showed that SGLD hits the approximate local minimum of the population risk in polynomial time. Mangoubi and Vishnoi [2018] further demonstrated SGLD with simulated annealing has a higher chance to obtain the global minima on a wider class of non-convex functions. However, all the analyses fail when DNN has too many parameters, and the over-specified model tends to have a large prediction variance, resulting in poor generalization and causing over-fitting. Therefore, a proper model selection is on demand at this situation.

A standard method to deal with model selection is variable selection. Notably, the best variable selection based on the $L_0$ penalty is conceptually ideal for sparsity detection but is computationally slow. Two alternatives emerged to approximate it. On the one hand, penalized likelihood approaches, such as Lasso [Tibshirani, 1994], induce sparsity due to the geometry that underlies the $L_1$ penalty. To better handle highly correlated variables, Elastic Net was proposed [Zou and Hastie, 2005] and makes a compromise between $L_1$ penalty and $L_2$ penalty. On the other hand, spike-and-slab approaches to Bayesian variable selection originates from probabilistic considerations. George and McCulloch [1993] proposed to build a continuous approximation of the spike-and-slab prior to sample from a hierarchical Bayesian model using Gibbs sampling. This continuous relaxation inspired the efficient EM variable selection (EMVS) algorithm in linear models [Ročková and George, 2014, 2018].

Despite the advances of model selection in linear systems, model selection in DNNs has received less attention. Ghosh et al. [2018] proposed to use variational inference (VI) based on regularized horseshoe priors to obtain a compact model. Liang et al. [2018] presented the theory of posterior consistency for Bayesian neural networks (BNNs) with Gaussian priors, and Ye and Sun [2018] applied a greedy elimination algorithm to conduct group model selection with the group Lasso penalty. Although these works only show the performance of shallow BNNs, the experimental methodologies imply the potential of model selection in DNNs. Louizos et al. [2017] studied scale mixtures of Gaussian priors and half-Cauchy scale priors for the hidden units of VGG models [Simonyan and Zisserman, 2014] and achieved good model compression performance on CIFAR10 [Krizhevsky, 2009] using VI. However, due to the limitation of VI in non-convex optimization, the compression is still not sparse enough and can be further optimized.

Over-parameterized DNNs often demand for tremendous memory use and heavy computational resources, which is impractical for smart devices. More critically, over-parametrization frequently overfits the data and results in worse performance [Lin et al., 2017]. To ensure the efficiency of the sparse sampling algorithm without over-shrinkage in DNN models, we propose an AEB method to adaptively sample from a hierarchical Bayesian DNN model with spike-and-slab Gaussian-Laplace (SSGL) priors and the priors are learned through optimization instead of sampling. The AEB method differs from the full Bayesian method in that the priors are inferred from the empirical data and the uncertainty of the priors is no longer considered to speed up the inference. In order to optimize the latent variables without affecting the convergence to the asymptotically correct distribution, stochastic approximation (SA) [Benveniste et al., 1990], a standard method for adaptive sampling [Andrieu et al., 2005, Liang, 2010], naturally fits to train the adaptive hierarchical Bayesian model. As a result, the asymptotic property allows us to combine simulated annealing and/or parallel tempering to accelerate the non-convex learning.

In this paper, we propose a sparse Bayesian deep learning algorithm, SG-MCMC-SA, to adaptively learn the hierarchical Bayes mixture models in DNNs. This algorithm has four main contributions:

- We propose a novel AEB method to efficiently train hierarchical Bayesian mixture DNN models, where the parameters are learned through sampling while the priors are learned through optimization.

- We prove the convergence of this approach to the asymptotically correct distribution, and it can be further generalized to a class of adaptive sampling algorithms for estimating state-space models in deep learning.

- We apply this adaptive sampling algorithm in the DNN compression problems firstly, with potential extension to a variety of model compression problems.

- It achieves the state of the art in terms of compression rates, which is 91.68% accuracy on CIFAR10 using only 27K parameters (90% sparsity) with Resnet20 [He et al., 2016].

## 2 Stochastic Gradient MCMC

We denote the set of model parameters by $\boldsymbol{\beta}$, the learning rate at time $k$ by $\epsilon^{(k)}$, the entire data by $\mathcal{D} = \{\boldsymbol{d}_i\}_{i=1}^N$, where $\boldsymbol{d}_i = (\boldsymbol{x}_i, y_i)$, the log of posterior by $L(\boldsymbol{\beta})$. The mini-batch of data $\mathcal{B}$ is of size $n$ with indices $\mathcal{S} = \{s_1, s_2, ..., s_n\}$, where $s_i \in \{1, 2, ..., N\}$. Stochastic gradient $\nabla_{\boldsymbol{\beta}} \tilde{L}(\boldsymbol{\beta})$ from a mini-batch of data $\mathcal{B}$ randomly sampled from $\mathcal{D}$ is used to approximate $\nabla_{\boldsymbol{\beta}} L(\boldsymbol{\beta})$:

$$\nabla_{\boldsymbol{\beta}} \tilde{L}(\boldsymbol{\beta}) = \nabla_{\boldsymbol{\beta}} \log \mathrm{P}(\boldsymbol{\beta}) + \frac{N}{n} \sum_{i \in \mathcal{S}} \nabla_{\boldsymbol{\beta}} \log \mathrm{P}(\boldsymbol{d}_i | \boldsymbol{\beta}). \tag{1}$$

SGLD (no momentum) is formulated as follows:

$$\boldsymbol{\beta}^{(k+1)} = \boldsymbol{\beta}^{(k)} + \epsilon^{(k)}\nabla_{\boldsymbol{\beta}}\tilde{L}(\boldsymbol{\beta}^{(k)}) + \mathcal{N}(0, 2\epsilon^{(k)}\tau^{-1}), \tag{2}$$

where $\tau > 0$ denotes the inverse temperature. It has been shown that SGLD asymptotically converges to a stationary distribution $\pi(\boldsymbol{\beta}|\mathcal{D}) \propto e^{\tau L(\boldsymbol{\beta})}$ [Teh et al., 2016, Zhang et al., 2017]. As $\tau$ increases and $\epsilon$ decreases gradually, the solution tends towards the global optima with a higher probability. Another variant of SG-MCMC, SGHMC [Chen et al., 2014, Ma et al., 2015], proposes to generate samples as follows:

$$\begin{cases} d\boldsymbol{\beta} = \boldsymbol{r}dt, \\ d\boldsymbol{r} = \nabla_{\boldsymbol{\beta}}\tilde{L}(\boldsymbol{\beta})dt - \boldsymbol{C}\boldsymbol{r}dt + \mathcal{N}(0, 2\boldsymbol{B}\tau^{-1}dt) + \mathcal{N}(0, 2(\boldsymbol{C} - \hat{\boldsymbol{B}})\tau^{-1}dt), \end{cases} \tag{3}$$

where $\boldsymbol{r}$ is the momentum item, $\hat{\boldsymbol{B}}$ is an estimate of the stochastic gradient variance, $\boldsymbol{C}$ is a user-specified friction term. Regarding the discretization of (3), we follow the numerical method proposed by Saatci and Wilson [2017] due to its convenience to import parameter settings from SGD.

## 3 Empirical Bayesian via Stochastic Approximation

### 3.1 A hierarchical formulation with deep SSGL priors

Inspired by the hierarchical Bayesian formulation for sparse inference [George and McCulloch, 1993], we assume the weight $\boldsymbol{\beta}_{lj}$ in sparse layer $l$ with index $j$ follows the SSGL prior

$$\beta_{lj}|\sigma^2, \gamma_{lj} \sim (1 - \gamma_{lj})\mathcal{L}(0, \sigma v_0) + \gamma_{lj}\mathcal{N}(0, \sigma^2 v_1). \tag{4}$$

where $\gamma_{lj} \in \{0,1\}$, $\boldsymbol{\beta}_l \in \mathbb{R}^{p_l}$, $\sigma^2 \in \mathbb{R}$, $\mathcal{L}(0, \sigma v_0)$ denotes a Laplace distribution with mean 0 and scale $\sigma v_0$, and $\mathcal{N}(0, \sigma^2 v_1)$ denotes a Normal distribution with mean 0 and variance $\sigma^2 v_1$. The sparse layer can be the fully connected layers (FC) in a shallow CNN or Convolutional layers in ResNet. If we have $\gamma_{lj} = 0$, the prior behaves like Lasso, which leads to a shrinkage effect; when $\gamma_{lj} = 1$, the $L_2$ penalty dominates. The likelihood follows

$$\pi(\mathcal{B}|\boldsymbol{\beta}, \sigma^2) = \begin{cases} \dfrac{\exp\left\{-\dfrac{\sum_{i \in \mathcal{S}}(y_i - \psi(\boldsymbol{x}_i; \boldsymbol{\beta}))^2}{2\sigma^2}\right\}}{(2\pi\sigma^2)^{n/2}} & \text{(regression)}, \\ \displaystyle\prod_{i \in \mathcal{S}} \dfrac{\exp\{\psi_{y_i}(\boldsymbol{x}_i; \boldsymbol{\beta})\}}{\sum_{t=1}^{K} \exp\{\psi_t(\boldsymbol{x}_i; \boldsymbol{\beta})\}} & \text{(classification)}, \end{cases} \tag{5}$$

where $\psi(\boldsymbol{x}_i; \boldsymbol{\beta})$ is a linear or non-linear mapping, and $y_i \in \{1, 2, \ldots, K\}$ is the response value of the $i$-th example. In addition, the variance $\sigma^2$ follows an inverse gamma prior $\pi(\sigma^2) = IG(\nu/2, \nu\lambda/2)$. The i.i.d. Bernoulli prior is used for $\boldsymbol{\gamma}$, namely $\pi(\gamma_l|\delta_l) = \delta_l^{|\gamma_l|}(1 - \delta_l)^{p_l - |\gamma_l|}$ where $\delta_l \in \mathbb{R}$ follows Beta distribution $\pi(\delta_l) \propto \delta_l^{a-1}(1 - \delta_l)^{b-1}$. The use of self-adaptive penalty enables the model to learn the level of sparsity automatically. Finally, our posterior follows

$$\pi(\boldsymbol{\beta}, \sigma^2, \delta, \boldsymbol{\gamma}|\mathcal{B}) \propto \pi(\mathcal{B}|\boldsymbol{\beta}, \sigma^2)^{\frac{N}{n}} \pi(\boldsymbol{\beta}|\sigma^2, \boldsymbol{\gamma})\pi(\sigma^2|\boldsymbol{\gamma})\pi(\boldsymbol{\gamma}|\delta)\pi(\delta). \tag{6}$$

### 3.2 Empirical Bayesian with approximate priors

To speed up the inference, we propose the AEB method by sampling $\boldsymbol{\beta}$ and optimizing $\sigma^2, \delta, \boldsymbol{\gamma}$, where uncertainty of the hyperparameters are not considered. Because the binary variable $\boldsymbol{\gamma}$ is harder to optimize directly, we consider optimizing the adaptive posterior $\mathbb{E}_{\boldsymbol{\gamma}|\cdot,\mathcal{D}}\left[\pi(\boldsymbol{\beta}, \sigma^2, \delta, \boldsymbol{\gamma}|\mathcal{D})\right]$ * instead. Due to the limited memory, which restricts us from sampling directly from $\mathcal{D}$, we choose to sample $\boldsymbol{\beta}$ from $\mathbb{E}_{\boldsymbol{\gamma}|\cdot,\mathcal{D}}\left[\mathbb{E}_{\mathcal{B}}\left[\pi(\boldsymbol{\beta}, \sigma^2, \delta, \boldsymbol{\gamma}|\mathcal{B})\right]\right]$ †. By Fubini's theorem and Jensen's inequality, we have

$$\begin{aligned} \log \mathbb{E}_{\boldsymbol{\gamma}|\cdot,\mathcal{D}}\left[\mathbb{E}_{\mathcal{B}}\left[\pi(\boldsymbol{\beta}, \sigma^2, \delta, \boldsymbol{\gamma}|\mathcal{B})\right]\right] &= \log \mathbb{E}_{\mathcal{B}}\left[\mathbb{E}_{\boldsymbol{\gamma}|\cdot,\mathcal{D}}\left[\pi(\boldsymbol{\beta}, \sigma^2, \delta, \boldsymbol{\gamma}|\mathcal{B})\right]\right] \\ &\geq \mathbb{E}_{\mathcal{B}}\left[\log \mathbb{E}_{\boldsymbol{\gamma}|\cdot,\mathcal{D}}\left[\pi(\boldsymbol{\beta}, \sigma^2, \delta, \boldsymbol{\gamma}|\mathcal{B})\right]\right] \geq \mathbb{E}_{\mathcal{B}}\left[\mathbb{E}_{\boldsymbol{\gamma}|\cdot,\mathcal{D}}\left[\log \pi(\boldsymbol{\beta}, \sigma^2, \delta, \boldsymbol{\gamma}|\mathcal{B})\right]\right]. \end{aligned} \tag{7}$$

Instead of tackling $\pi(\boldsymbol{\beta}, \sigma^2, \delta, \boldsymbol{\gamma}|\mathcal{D})$ directly, we propose to iteratively update the lower bound $Q$

$$Q(\boldsymbol{\beta}, \sigma, \delta|\boldsymbol{\beta}^{(k)}, \sigma^{(k)}, \delta^{(k)}) = \mathbb{E}_{\mathcal{B}}\left[\mathbb{E}_{\boldsymbol{\gamma}|\mathcal{D}}\left[\log \pi(\boldsymbol{\beta}, \sigma^2, \delta, \boldsymbol{\gamma}|\mathcal{B})\right]\right]. \tag{8}$$

Given $(\boldsymbol{\beta}^{(k)}, \sigma^{(k)}, \delta^{(k)})$ at the k-th iteration, we first sample $\boldsymbol{\beta}^{(k+1)}$ from $Q$, then optimize $Q$ with respect to $\sigma, \delta$ and $\mathbb{E}_{\boldsymbol{\gamma}_l|\cdot,\mathcal{D}}$ via SA, where $\mathbb{E}_{\boldsymbol{\gamma}_l|\cdot,\mathcal{D}}$ is used since $\boldsymbol{\gamma}$ is treated as unobserved variable. To make the computation easier, we decompose our $Q$ as follows:

$$Q(\boldsymbol{\beta}, \sigma, \delta|\boldsymbol{\beta}^{(k)}, \sigma^{(k)}, \delta^{(k)}) = Q_1(\beta, \sigma|\boldsymbol{\beta}^{(k)}, \sigma^{(k)}, \delta^{(k)}) + Q_2(\delta|\boldsymbol{\beta}^{(k)}, \sigma^{(k)}, \delta^{(k)}) + C, \tag{9}$$

Denote $\mathcal{X}$ and $\mathcal{C}$ as the sets of the indices of sparse and non-sparse layers, respectively. We have:

$$Q_1(\boldsymbol{\beta}|\boldsymbol{\beta}^{(k)}, \sigma^{(k)}, \delta^{(k)}) = \underbrace{\frac{N}{n} \log \pi(\mathcal{B}|\boldsymbol{\beta})}_{\text{log likelihood}} - \underbrace{\sum_{l \in \mathcal{C}} \sum_{j \in p_l} \frac{\beta_{lj}^2}{2\sigma_0^2}}_{\text{non-sparse layers } \mathcal{C}} - \frac{p+\nu+2}{2} \log(\sigma^2)$$

$$-\sum_{l \in \mathcal{X}} \sum_{j \in p_l} \underbrace{\left[\frac{|\beta_{lj}| \overbrace{\mathbb{E}_{\boldsymbol{\gamma}_l|\cdot,\mathcal{D}}\left[\frac{1}{v_0(1-\gamma_{lj})}\right]}^{\kappa_{lj0}}}{\sigma} + \frac{\beta_{lj}^2 \overbrace{\mathbb{E}_{\boldsymbol{\gamma}_l|\cdot,\mathcal{D}}\left[\frac{1}{v_1\gamma_{lj}}\right]}^{\kappa_{lj1}}}{2\sigma^2}\right]}_{\text{deep SSGL priors in sparse layers } \mathcal{X}} - \frac{\nu\lambda}{2\sigma^2} \tag{10}$$

$$Q_2(\delta_l|\boldsymbol{\beta}_l^{(k)}, \delta_l^{(k)}) = \sum_{l \in \mathcal{X}} \sum_{j \in p_l} \log\left(\frac{\delta_l}{1-\delta_l}\right) \overbrace{\mathbb{E}_{\boldsymbol{\gamma}_l|\cdot,\mathcal{D}}\gamma_{lj}}^{\rho_{lj}} + (a-1)\log(\delta_l) + (p_l + b - 1)\log(1-\delta_l), \tag{11}$$

where $\rho, \kappa, \sigma$ and $\delta$ are to be estimated in the next section.

### 3.3 Empirical Bayesian via stochastic approximation

To simplify the notation, we denote the vector $(\rho, \kappa, \sigma, \delta)$ by $\boldsymbol{\theta}$. Our interest is to obtain the optimal $\boldsymbol{\theta}_*$ based on the asymptotically correct distribution $\pi(\boldsymbol{\beta}, \boldsymbol{\theta}_*)$. This implies that we need to obtain an estimate $\boldsymbol{\theta}_*$ that solves a fixed-point formulation $\int g_{\boldsymbol{\theta}_*}(\boldsymbol{\beta})\pi(\boldsymbol{\beta}, \boldsymbol{\theta}_*)d\boldsymbol{\beta} = \boldsymbol{\theta}_*$ [Shimkin, 2011], where $g_{\boldsymbol{\theta}}(\boldsymbol{\beta})$ is inspired by EMVS to obtain the optimal $\boldsymbol{\theta}$ based on the current $\boldsymbol{\beta}$. Define the random output $g_{\boldsymbol{\theta}}(\boldsymbol{\beta}) - \boldsymbol{\theta}$ as $H(\boldsymbol{\beta}, \boldsymbol{\theta})$ and the mean field function $h(\boldsymbol{\theta}) := \mathbb{E}[H(\boldsymbol{\beta}, \boldsymbol{\theta})]$. The stochastic approximation algorithm can be used to solve the fixed-point iterations:

(1) Sample $\boldsymbol{\beta}^{(k+1)}$ from a transition kernel $\Pi_{\boldsymbol{\theta}^{(k)}}(\boldsymbol{\beta})$, which yields the distribution $\pi(\boldsymbol{\beta}, \boldsymbol{\theta}^{(k)})$,

(2) Update $\boldsymbol{\theta}^{(k+1)} = \boldsymbol{\theta}^{(k)} + \omega^{(k+1)} H(\boldsymbol{\theta}^{(k)}, \boldsymbol{\beta}^{(k+1)}) = \boldsymbol{\theta}^{(k)} + \omega^{(k+1)}(h(\boldsymbol{\theta}^{(k)}) + \Omega^{(k)})$.

where $\omega^{(k+1)}$ is the step size. The equilibrium point $\boldsymbol{\theta}_*$ is obtained when the distribution of $\boldsymbol{\beta}$ converges to the invariant distribution $\pi(\boldsymbol{\beta}, \boldsymbol{\theta}_*)$. The stochastic approximation [Benveniste et al., 1990] differs from the Robbins–Monro algorithm in that sampling $\boldsymbol{\beta}$ from a transition kernel instead of a distribution introduces a Markov state-dependent noise $\Omega^{(k)}$ [Andrieu et al., 2005]. In addition, since variational technique is only used to approximate the priors, and the exact likelihood doesn't change, the algorithm falls into a class of adaptive SG-MCMC instead of variational inference.

Regarding the updates of $g_{\boldsymbol{\theta}}(\boldsymbol{\beta})$ with respect to $\rho$, we denote the optimal $\rho$ based on the current $\boldsymbol{\beta}$ and $\delta$ by $\tilde{\rho}$. We have that $\tilde{\rho}_{lj}^{(k+1)}$, the probability of $\beta_{lj}$ being dominated by the $L_2$ penalty is

$$\tilde{\rho}_{lj}^{(k+1)} = \mathbb{E}_{\boldsymbol{\gamma}_l|\cdot,\mathcal{B}}\gamma_{lj} = \mathrm{P}(\gamma_{lj} = 1|\boldsymbol{\beta}_l^{(k)}, \delta_l^{(k)}) = \frac{a_{lj}}{a_{lj} + b_{lj}}, \tag{12}$$

where $a_{lj} = \pi(\beta_{lj}^{(k)}|\gamma_{lj} = 1)\mathrm{P}(\gamma_{lj} = 1|\delta_l^{(k)})$ and $b_{lj} = \pi(\beta_{lj}^{(k)}|\gamma_{lj} = 0)\mathrm{P}(\gamma_{lj} = 0|\delta_l^{(k)})$. The choice of Bernoulli prior enables us to use $\mathrm{P}(\gamma_{lj} = 1|\delta_l^{(k)}) = \delta_l^{(k)}$.

Similarly, as to $g_{\boldsymbol{\theta}}(\boldsymbol{\beta})$ w.r.t. $\kappa$, the optimal $\tilde{\kappa}_{lj0}$ and $\tilde{\kappa}_{lj1}$ based on the current $\rho_{lj}$ are given by:

$$\tilde{\kappa}_{lj0} = \mathbb{E}_{\boldsymbol{\gamma}_l|\cdot,\mathcal{B}}\left[\frac{1}{v_0(1-\gamma_{lj})}\right] = \frac{1-\rho_{lj}}{v_0}; \quad \tilde{\kappa}_{lj1} = \mathbb{E}_{\boldsymbol{\gamma}_l|\cdot,\mathcal{B}}\left[\frac{1}{v_1\gamma_{lj}}\right] = \frac{\rho_{lj}}{v_1}. \tag{13}$$

To optimize $Q_1$ with respect to $\sigma$, by denoting $\text{diag}\{\kappa_{0li}\}_{i=1}^{p_l}$ as $\boldsymbol{\mathcal{V}}_{0l}$, $\text{diag}\{\kappa_{1li}\}_{i=1}^{p_l}$ as $\boldsymbol{\mathcal{V}}_{1l}$ we have:

$$\tilde{\sigma}^{(k+1)} = \begin{cases} \dfrac{R_b + \sqrt{R_b^2 + 4R_a R_c}}{2R_a} & \text{(regression)}, \\[3mm] \dfrac{C_b + \sqrt{C_b^2 + 4C_a C_c}}{2C_a} & \text{(classification)}, \end{cases} \tag{14}$$

where $R_a = N + \sum_{l \in \mathcal{X}} p_l + \nu$, $C_a = \sum_{l \in \mathcal{X}} p_l + \nu + 2$, $R_b = C_b = \sum_{l \in \mathcal{X}} ||\boldsymbol{\mathcal{V}}_{0l}\boldsymbol{\beta}_l^{(k+1)}||_1$, $R_c = I + J + \nu\lambda$, $C_c = J + \nu\lambda$, $I = \frac{N}{n}\sum_{i \in \mathcal{S}}\left(y_i - \psi(\boldsymbol{x}_i; \boldsymbol{\beta}^{(k+1)})\right)^2$, $J = \sum_{l \in \mathcal{X}} ||\boldsymbol{\mathcal{V}}_{1l}^{1/2}\boldsymbol{\beta}_l^{(k+1)}||^2$.[†]

To optimize $Q_2$, a closed-form update can be derived from Eq.(11) and Eq.(12) given batch data $\mathcal{B}$:

$$\tilde{\delta}_l^{(k+1)} = \underset{\delta_l \in \mathbb{R}}{\arg\max}\, Q_2(\delta_l | \boldsymbol{\beta}_l^{(k)}, \delta_l^{(k)}) = \frac{\sum_{j=1}^{p_l} \rho_{lj} + a - 1}{a + b + p_l - 2}. \tag{15}$$

### 3.4 Pruning strategy

There are quite a few methods for pruning neural networks including the oracle pruning and the easy-to-use magnitude-based pruning [Molchanov et al., 2017]. Although the magnitude-based unit pruning shows more computational savings [Gomez et al., 2018], it doesn't demonstrate robustness under coarser pruning [Han et al., 2016, Gomez et al., 2018]. Pruning based on the probability $\rho$ is also popular in the Bayesian community, but achieving the target sparsity in sophisticated networks requires extra fine-tuning. We instead apply the magnitude-based weight-pruning to our Resnet compression experiments and refer to it as SGLD-SA, which is detailed in Algorithm 1. The corresponding variant of SGHMC with SA is referred to as SGHMC-SA.

## 4 Convergence Analysis

The key to guaranteeing the convergence of the adaptive SGLD algorithm is to use Poisson's equation to analyze additive functionals. By decomposing the Markov state-dependent noise $\Omega$ into martingale difference sequences and perturbations, where the latter can be controlled by the regularity of the solution of Poisson's equation, we can guarantee the consistency of the latent variable estimators.

**Theorem 1** ($L_2$ convergence rate). *For any $\alpha \in (0, 1]$, under assumptions in Appendix B.1, the algorithm satisfies: there exists a constant $\lambda$ and an optimum $\boldsymbol{\theta}^*$ such that*

$$\mathbb{E}\left[||\boldsymbol{\theta}^{(k)} - \boldsymbol{\theta}^*||^2\right] \leq \lambda k^{-\alpha}.$$

SGLD with adaptive latent variables forms a sequence of inhomogenous Markov chains and the weak convergence of $\boldsymbol{\beta}$ to the target posterior is equivalent to proving the weak convergence of SGLD with biased estimations of gradients. Inspired by Chen et al. [2015], we have:

**Corollary 1.** *Under assumptions in Appendix B.2, the random vector $\boldsymbol{\beta}^{(k)}$ from the adaptive transition kernel $\Pi_{\boldsymbol{\theta}^{(k-1)}}$ converges weakly to the invariant distribution $e^{\tau L(\boldsymbol{\beta}, \boldsymbol{\theta}^*)}$ as $\epsilon \to 0$ and $k \to \infty$.*

The smooth optimization of the priors makes the algorithm robust to bad initialization and avoids entrapment in poor local optima. In addition, the convergence to the asymptotically correct distribution allows us to combine simulated annealing to obtain better point estimates in non-convex optimization.

## 5 Experiments

### 5.1 Simulation of Large-p-Small-n Regression

We conduct the linear regression experiments with a dataset containing $n = 100$ observations and $p = 1000$ predictors. $\mathcal{N}_p(0, \boldsymbol{\Sigma})$ is chosen to simulate the predictor values $\boldsymbol{X}$ (training set) where $\boldsymbol{\Sigma} = (\Sigma)_{i,j=1}^p$ with $\Sigma_{i,j} = 0.6^{|i-j|}$. Response values $\boldsymbol{y}$ are generated from $\boldsymbol{X}\boldsymbol{\beta} + \boldsymbol{\eta}$, where $\boldsymbol{\beta} = (\beta_1, \beta_2, \beta_3, 0, 0, ..., 0)'$ and $\boldsymbol{\eta} \sim \mathcal{N}_n(\boldsymbol{0}, 3\boldsymbol{I}_n)$. We assume $\beta_1 \sim \mathcal{N}(3, \sigma_c^2)$, $\beta_2 \sim \mathcal{N}(2, \sigma_c^2)$,

---

[†]The quadratic equation has only one unique positive root. $||\cdot||$ refers to $L_2$ norm, $||\cdot||_1$ represents $L_1$ norm.

---

**Algorithm 1** SGLD-SA with SSGL priors

---

**Initialize:** $\boldsymbol{\beta}^{(1)}, \boldsymbol{\rho}^{(1)}, \boldsymbol{\kappa}^{(1)}, \sigma^{(1)}$ and $\delta^{(1)}$ from scratch, set target sparse rates $\mathbb{D}, \mho$ and $\mathbb{S}$
**for** $k \leftarrow 1 : k_{\max}$ **do**
    **Sampling**
    $\boldsymbol{\beta}^{(k+1)} \leftarrow \boldsymbol{\beta}^{(k)} + \epsilon^{(k)} \nabla_{\boldsymbol{\beta}} Q(\cdot | \mathcal{B}^{(k)}) + \mathcal{N}(0, 2\epsilon^{(k)}\tau^{-1})$
    **Stochastic Approximation for Latent Variables**
    **SA**: $\boldsymbol{\rho}^{(k+1)} \leftarrow (1 - \omega^{(k+1)})\boldsymbol{\rho}^{(k)} + \omega^{(k+1)}\tilde{\boldsymbol{\rho}}^{(k+1)}$ following Eq.(12)
    **SA**: $\boldsymbol{\kappa}^{(k+1)} \leftarrow (1 - \omega^{(k+1)})\boldsymbol{\kappa}^{(k)} + \omega^{(k+1)}\tilde{\boldsymbol{\kappa}}^{(k+1)}$ following Eq.(13)
    **SA**: $\sigma^{(k+1)} \leftarrow (1 - \omega^{(k+1)})\sigma^{(k)} + \omega^{(k+1)}\tilde{\sigma}^{(k+1)}$ following Eq.(14)
    **SA**: $\delta^{(k+1)} \leftarrow (1 - \omega^{(k+1)})\delta^{(k)} + \omega^{(k+1)}\tilde{\delta}^{(k+1)}$ following Eq.(15)
    **if Pruning then**
        Prune the bottom-$s\%$ lowest magnitude weights
        Increase the sparse rate $s \leftarrow \mathbb{S}(1 - \mathbb{D}^{k/\mho})$
    **end if**
**end for**

---

Table 1: Predictive errors in linear regression based on a test set considering different $v_0$ and $\sigma$

| MAE / MSE | $v_0$=0.01, $\sigma$=2 | $v_0$=0.1, $\sigma$=2 | $v_0$=0.01, $\sigma$=1 | $v_0$=0.1, $\sigma$=1 |
|---|---|---|---|---|
| SGLD-SA | **1.89 / 5.56** | **1.72 / 5.64** | **1.48 / 3.51** | **1.54 / 4.42** |
| SGLD-EM | 3.49 / 19.31 | 2.23 / 8.22 | 2.23 / 19.28 | 2.07 / 6.94 |
| SGLD | 15.85 / 416.39 | 15.85 / 416.39 | 11.86 / 229.38 | 7.72 / 88.90 |

$\beta_3 \sim \mathcal{N}(1, \sigma_c^2)$, $\sigma_c = 0.2$. We introduce some hyperparameters, but most of them are uninformative. We fix $\tau = 1, \lambda = 1, \nu = 1, v_1 = 10, \delta = 0.5, b = p$ and set $a = 1$. The learning rate follows $\epsilon^{(k)} = 0.001 \times k^{-\frac{1}{3}}$, and the step size is given by $\omega^{(k)} = 10 \times (k + 1000)^{-0.7}$. We vary $v_0$ and $\sigma$ to show the robustness of SGLD-SA to different initializations. In addition, to show the superiority of the adaptive update, we compare SGLD-SA with the intuitive implementation of the EMVS to SGLD and refer to this algorithm as SGLD-EM, which is equivalent to setting $\omega^{(k)} := 1$ in SGLD-SA. To obtain the stochastic gradient, we randomly select 50 observations and calculate the numerical gradient. SGLD is sampled from the same hierarchical model without updating the latent variables.

We simulate $500,000$ samples from the posterior distribution, and also simulate a test set with 50 observations to evaluate the prediction. As shown in Fig.1 (d), all three algorithms are fitted very well in the training set, however, SGLD fails completely in the test set (Fig.1 (e)), indicating the over-fitting problem of SGLD without proper regularization when the latent variables are not updated. Fig.1 (f) shows that although SGLD-EM successfully identifies the right variables, the estimations are lower biased. The reason is that SGLD-EM fails to regulate the right variables with $L_2$ penalty, and $L_1$ leads to a greater amount of shrinkage for $\boldsymbol{\beta}_1, \boldsymbol{\beta}_2$ and $\boldsymbol{\beta}_3$ (Fig. 1 (a-c)), implying the importance of the adaptive update via SA in the stochastic optimization of the latent variables. In addition, from Fig. 1(a), Fig. 1(b) and Fig.1(c), we see that SGLD-SA is the only algorithm among the three that quantifies the uncertainties of $\beta_1$, $\beta_2$ and $\beta_3$ and always gives the best prediction as shown in Table.1. We notice that SGLD-SA is fairly robust to various hyperparameters.

For the simulation of SGLD-SA in logistic regression and the evaluation of SGLD-SA on UCI datasets, we leave the results in Appendix C and D.

## 5.2 Classification with Auto-tuning Hyperparameters

The following experiments are based on non-pruning SG-MCMC-SA, the goal is to show that auto-tuning sparse priors are useful to avoid over-fitting. The posterior average is applied to each Bayesian model. We implement all the algorithms in Pytorch [Paszke et al., 2017]. The first DNN is a standard 2-Conv-2-FC CNN model of 670K parameters (see details in Appendix D.1).

The first set of experiments is to compare methods on the same model without using data augmentation (DA) and batch normalization (BN) [Ioffe and Szegedy, 2015]. We refer to the general CNN without dropout as Vanilla, with 50% dropout rate applied to the hidden units next to FC1 as Dropout.

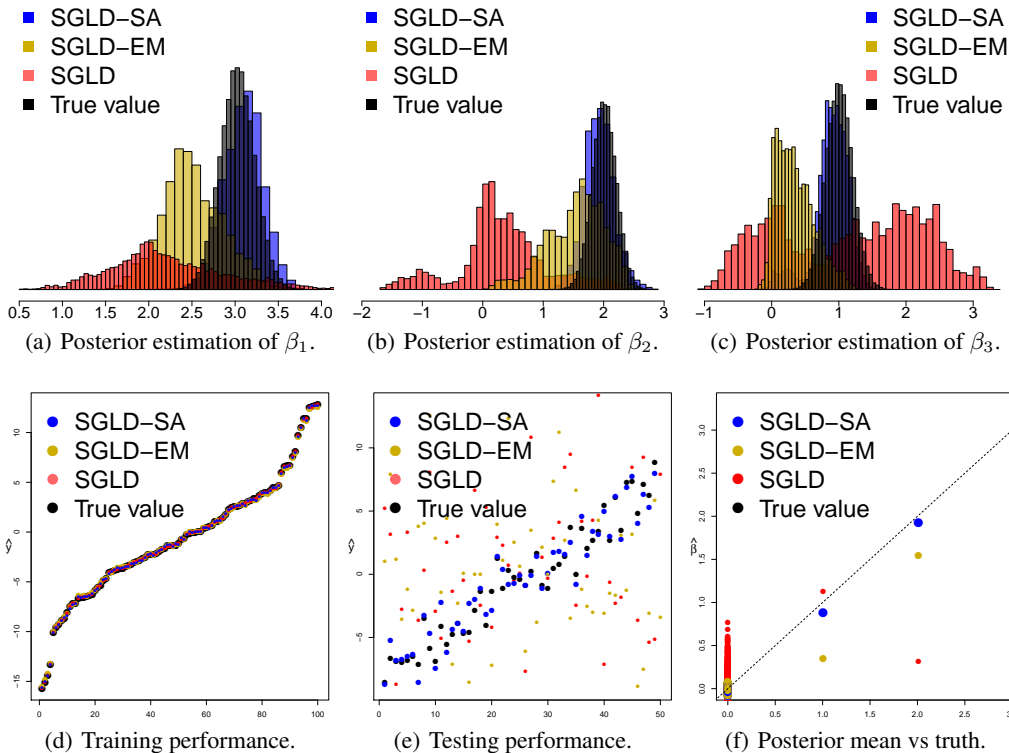

(a) Posterior estimation of $\beta_1$.     (b) Posterior estimation of $\beta_2$.     (c) Posterior estimation of $\beta_3$.

(d) Training performance.     (e) Testing performance.     (f) Posterior mean vs truth.

Figure 1: Linear regression simulation when $v_0 = 0.1$ and $\sigma = 1$.

Vanilla and Dropout models are trained with Adam [Kingma and Ba, 2014] and Pytorch default parameters (with learning rate 0.001). We use SGHMC as a benchmark method as it is also sampling-based and has a close relationship with the popular momentum based optimization approaches in DNNs. SGHMC-SA differs from SGHMC in that SGHMC-SA keeps updating SSGL priors for the first FC layer while they are fixed in SGHMC. We set the training batch size $n = 1000$, $a, b = p$ and $\nu, \lambda = 1000$. The hyperparameters for SGHMC-SA are set to $v_0 = 1, v_1 = 0.1$ and $\sigma = 1$ to regularize the over-fitted space. The learning rate is set to $5 \times 10^{-7}$, and the step size is $\omega^{(k)} = 1 \times (k + 1000)^{-\frac{3}{4}}$. We use a thinning factor 500 to avoid a cumbersome system. Fixed temperature can also be powerful in escaping "shallow" local traps [Zhang et al., 2017], our temperatures are set to $\tau = 1000$ for MNIST and $\tau = 2500$ for FMNIST.

The four CNN models are tested on MNIST and Fashion MNIST (FMNIST) [Xiao et al., 2017] dataset. Performance of these models is shown in Tab.2. Compared with SGHMC, our SGHMC-SA outperforms SGHMC on both datasets. We notice the posterior averages from SGHMC-SA and SGHMC obtain much better performance than Vanilla and Dropout. Without using either DA or BN, SGHMC-SA achieves 99.59% which outperforms some state-of-the-art models, such as Maxout Network (99.55%) [Goodfellow et al., 2013] and pSGLD (99.55%) [Li et al., 2016] . In F-MNIST, SGHMC-SA obtains 93.01% accuracy, outperforming all other competing models.

To further test the performance, we apply DA and BN to the following experiments (see details in Appendix D.2) and refer to the datasets as DA-MNIST and DA-FMNIST. All the experiments are conducted using a 2-Conv-BN-3-FC CNN of 490K parameters. Using this model, we obtain the state-of-the-art 99.75% on DA-MNIST (200 epochs) and 94.38% on DA-FMNIST (1000 epochs) as shown in Tab. 2. The results are noticeable, because posterior average is only conducted on a single shallow CNN.

## 5.3  Defenses against Adversarial Attacks

Continuing with the setup in Sec. 5.2, the third set of experiments focuses on evaluating model robustness. We apply the *Fast Gradient Sign* method [Goodfellow et al., 2014] to generate the

Table 2: Classification accuracy using shallow networks

| DATASET | MNIST | DA-MNIST | FMNIST | DA-FMNIST |
|---|---|---|---|---|
| VANILLA | 99.31 | 99.54 | 92.73 | 93.14 |
| DROPOUT | 99.38 | 99.56 | 92.81 | 93.35 |
| SGHMC | 99.47 | 99.63 | 92.88 | 94.29 |
| **SGHMC-SA** | **99.59** | **99.75** | **93.01** | **94.38** |

adversarial examples with one single gradient step as in Papernot et al. [2016]'s study:

$$\boldsymbol{x}_{adv} \leftarrow \boldsymbol{x} - \zeta \cdot \text{sign}\{\delta_{\boldsymbol{x}} \max_y \log \text{P}(y\,|\boldsymbol{x})\},$$

where $\zeta$ ranges from $0.1, 0.2, \ldots, 0.5$ to control the different levels of adversarial attacks.

Similar to the setup in Li and Gal [2017], we normalize the adversarial images by clipping to the range $[0, 1]$. In Fig. 2(b) and Fig.2(d), we see no significant difference among all the four models in the early phase. As the degree of adversarial attacks arises, the images become vaguer as shown in Fig.2(a) and Fig.2(c). The performance of Vanilla decreases rapidly, reflecting its poor defense against adversarial attacks, while Dropout performs better than Vanilla. But Dropout is still significantly worse than the sampling based methods. The advantage of SGHMC-SA over SGHMC becomes more significant when $\zeta > 0.25$. In the case of $\zeta = 0.5$ in MNIST where the images are hardly recognizable, both Vanilla and Dropout models fail to identify the right images and their predictions are as worse as random guesses. However, SGHMC-SA model achieves roughly 11% higher than these two models and 1% higher than SGHMC, which demonstrates the robustness of SGHMC-SA.

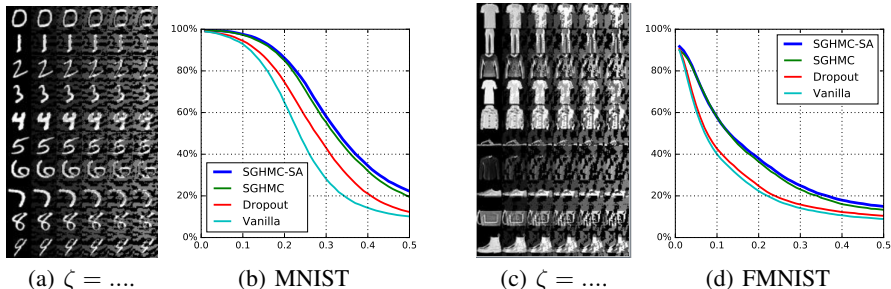

(a) $\zeta = \ldots$    (b) MNIST    (c) $\zeta = \ldots$    (d) FMNIST

Figure 2: Adversarial test accuracies based on adversarial images of different levels

## 5.4 Residual Network Compression

Our compression experiments are conducted on the CIFAR-10 dataset [Krizhevsky, 2009] with DA. SGHMC and the non-adaptive SGHMC-EM are chosen as baselines. Simulated annealing is used to enhance the non-convex optimization and the methods with simulated annealing are referred to as A-SGHMC, A-SGHMC-EM and A-SGHMC-SA, respectively. We report the best point estimate.

We first use SGHMC to train a Resnet20 model and apply the magnitude-based criterion to prune weights to all convolutional layers (except the very first one). All the following methods are evaluated based on the same setup except for different step sizes to learn the latent variables. The sparse training takes 1000 epochs. The mini-batch size is 1000. The learning rate starts from 2e-9 [†] and is divided by 10 at the 700th and 900th epoch. We set the inverse temperature $\tau$ to 1000 and multiply $\tau$ by 1.005 every epoch . We fix $\nu = 1000$ and $\lambda = 1000$ for the inverse gamma prior. $v_0$ and $v_1$ are tuned based on different sparsity to maximize the performance. The smooth increase of the sparse rate follows the pruning rule in Algorithm 1, and $\mathbb{D}$ and $\mho$ are set to 0.99 and 50, respectively. The increase in the sparse rate $s$ is faster in the beginning and slower in the later phase to avoid destroying the network structure. Weight decay in the non-sparse layers $\mathcal{C}$ is set as 25.

As shown in Table 3, A-SGHMC-SA doesn't distinguish itself from A-SGHMC-EM and A-SGHMC when the sparse rate $\mathbb{S}$ is small, but outperforms the baselines given a large sparse rate. The pretrained model has accuracy 93.90%, however,the prediction performance can be improved to the state-of-the-art 94.27% with 50% sparsity. Most notably, we obtain 91.68% accuracy based on 27K parameters

---

[†]It is equivalent to setting the learning rate to 1e-4 when we don't multiply the likelihood with $\frac{N}{n}$.

Table 3: Resnet20 Compression on CIFAR10. When $\mathbb{S} = 0.9$, we fix $v_0 = 0.005$, $v_1 =$1e-5; When $\mathbb{S} = 0.7$, we fix $v_0 = 0.1$, $v_1 =$5e-5; When $\mathbb{S} = 0.5$, we fix $v_0 = 0.1$, $v_1 =$5e-4; When $\mathbb{S} = 0.3$, we fix $v_0 = 0.5$, $v_1 =$1e-3.

| METHODS \ $\mathbb{S}$ | 30% | 50% | 70% | 90% |
|---|---|---|---|---|
| A-SGHMC | 94.07 | 94.16 | 93.16 | 90.59 |
| A-SGHMC-EM | 94.18 | 94.19 | 93.41 | 91.26 |
| SGHMC-SA | 94.13 | 94.11 | 93.52 | 91.45 |
| **A-SGHMC-SA** | **94.23** | **94.27** | **93.74** | **91.68** |

(90% sparsity) in Resnet20. By contrast, targeted dropout obtained 91.48% accuracy based on 47K parameters (90% sparsity) of Resnet32 [Gomez et al., 2018], BC-GHS achieves 91.0% accuracy based on 8M parameters (94.5% sparsity) of VGG models [Louizos et al., 2017]. We also notice that when simulated annealing is not used as in SGHMC-SA, the performance will decrease by 0.2% to 0.3%. When we use batch size 2000 and inverse temperature schedule $\tau^{(k)} = 20 \times 1.01^k$, A-SGHMC-SA still achieves roughly the same level, but the prediction of SGHMC-SA can be 1% lower than A-SGHMC-SA.

## 6   Conclusion

We propose a novel AEB method to adaptively sample from hierarchical Bayesian DNNs and optimize the spike-and-slab priors, which yields a class of scalable adaptive sampling algorithms in DNNs. We prove the convergence of this approach to the asymptotically correct distribution. By adaptively searching and penalizing the over-fitted parameters, the proposed method achieves higher prediction accuracy over the traditional SG-MCMC methods in both simulated examples and real applications and shows more robustness towards adversarial attacks. Together with the magnitude-based weight pruning strategy and simulated annealing, the AEB-based method, A-SGHMC-SA, obtains the state-of-the-art performance in model compression.

**Acknowledgments**

We would like to thank Prof. Vinayak Rao, Dr. Yunfan Li and the reviewers for their insightful comments. We acknowledge the support from the National Science Foundation (DMS-1555072, DMS-1736364, DMS-1821233 and DMS-1818674) and the GPU grant program from NVIDIA.

## Footnotes

*$\mathbb{E}_{\boldsymbol{\gamma}|\cdot,\mathcal{D}}[\cdot]$ is short for $\mathbb{E}_{\boldsymbol{\gamma}|\beta^{(k)},\sigma^{(k)},\delta^{(k)},\mathcal{D}}[\cdot]$.

†$\mathbb{E}_{\mathcal{B}}[\pi(\boldsymbol{\beta}, \sigma^2, \delta, \boldsymbol{\gamma}|\mathcal{B})]$ denotes $\int_{\mathcal{D}} \pi(\boldsymbol{\beta}, \sigma^2, \delta, \boldsymbol{\gamma}|\mathcal{B})d\mathcal{B}$

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
