[Supplementary Material]

# Supplimentary Material for *An Adaptive Empirical Bayesian Method for Sparse Deep Learning*

**Wei Deng**
Purdue University
deng106@purdue.edu

**Xiao Zhang**
Purdue University
zhang923@purdue.edu

**Faming Liang**
Purdue University
fmliang@purdue.edu

**Guang Lin**
Purdue University
guanglin@purdue.edu

In this supplementary material, we review the related methodologies in §1, prove the convergence in §2, present additional simulation of logistic regression in §3, illustrate more regression examples on UCI datasets in §4, and show the experimental setup in §5.

## 1 Stochastic Approximation

### 1.1 Special Case: Robbins–Monro Algorithm

Robbins–Monro algorithm is the first stochastic approximation algorithm to deal with the root finding problem which also applies to the stochastic optimization problem. Given the random output of $H(\boldsymbol{\theta}, \boldsymbol{\beta})$ with respect to $\boldsymbol{\beta}$, our goal is to find $\boldsymbol{\theta}^*$ such that

$$h(\boldsymbol{\theta}^*) = \mathbb{E}_{\boldsymbol{\theta}^*}[H(\boldsymbol{\theta}^*, \boldsymbol{\beta})] = \int H(\boldsymbol{\theta}^*, \boldsymbol{\beta}) f_{\boldsymbol{\theta}*}(d\boldsymbol{\beta}) = 0, \tag{1}$$

where $\mathbb{E}_{\boldsymbol{\theta}^*}$ denotes the expectation with respect to the distribution of $\boldsymbol{\beta}$ given $\boldsymbol{\theta}^*$. To implement the Robbins–Monro Algorithm, we can generate iterates as follows[*]:

(1) Sample $\boldsymbol{\beta}_{k+1}$ from the invariant distribution $f_{\boldsymbol{\theta}_k}(\boldsymbol{\beta})$,

(2) Update $\boldsymbol{\theta}_{k+1} = \boldsymbol{\theta}_k + \omega_{k+1} H(\boldsymbol{\theta}_k, \boldsymbol{\beta}_{k+1})$.

Note that in this algorithm, $H(\boldsymbol{\theta}, \boldsymbol{\beta})$ is the unbiased estimator of $h(\boldsymbol{\theta})$, that is for $k \in \mathrm{N}^+$, we have

$$\mathbb{E}_{\boldsymbol{\theta}_k}[H(\boldsymbol{\theta}_k, \boldsymbol{\beta}_{k+1}) - h(\boldsymbol{\theta}_k)|\mathcal{F}_k] = 0. \tag{2}$$

If there exists an antiderivative $Q(\boldsymbol{\theta}, \boldsymbol{\beta})$ that satisfies $H(\boldsymbol{\theta}, \boldsymbol{\beta}) = \nabla_{\boldsymbol{\theta}} Q(\boldsymbol{\theta}, \boldsymbol{\beta})$ and $E_{\boldsymbol{\theta}}[Q(\boldsymbol{\theta}, \boldsymbol{\beta})]$ is concave, it is equivalent to solving the stochastic optimization problem $\max_{\boldsymbol{\theta} \in \boldsymbol{\Theta}} E_{\boldsymbol{\theta}}[Q(\boldsymbol{\theta}, \boldsymbol{\beta})]$.

### 1.2 General Stochastic Approximation

The stochastic approximation algorithm is an iterative recursive algorithm consisting of two steps:

(1) Sample $\boldsymbol{\beta}_{k+1}$ from the transition kernel $\Pi_{\boldsymbol{\theta}_k}(\boldsymbol{\beta}_k, \cdot)$, which admits $f_{\boldsymbol{\theta}_k}(\boldsymbol{\beta})$ as the invariant distribution,

(2) Update $\boldsymbol{\theta}_{k+1} = \boldsymbol{\theta}_k + \omega_{k+1} H(\boldsymbol{\theta}_k, \boldsymbol{\beta}_{k+1})$.

The general stochastic approximation [Benveniste et al., 1990] differs from the Robbins-Monro algorithm in that sampling $x$ from a transition kernel instead of a distribution introduces a Markov state-dependent noise $H(\theta_k, x_{k+1}) - h(\theta_k)$.

---

[*]We change the notation a little bit, where $\boldsymbol{\beta}_k \in \mathbb{R}^d$ and $\boldsymbol{\theta}_k$ are the parameters at the $k$-th iteration.

## 2 Convergence Analysis

### 2.1 Convergence of Hidden Variables

The stochastic gradient Langevin Dynamics with a stochastic approximation adaptation (SGLD-SA) is a mixed half-optimization-half-sampling algorithm to handle complex Bayesian posterior with latent variables, e.g. the conjugate spike-slab hierarchical prior formulation. Each iteration of the algorithm consists of the following steps:

(1) Sample $\boldsymbol{\beta}_{k+1}$ using SGLD based on $\boldsymbol{\theta}_k$, i.e.

$$\boldsymbol{\beta}_{k+1} = \boldsymbol{\beta}_k + \epsilon \nabla_{\boldsymbol{\beta}} \tilde{L}(\boldsymbol{\beta}_k, \boldsymbol{\theta}_k) + \sqrt{2\epsilon\tau^{-1}}\boldsymbol{\eta}_k, \tag{3}$$

where $\boldsymbol{\eta}_k \sim \mathcal{N}(0, \boldsymbol{I})$;

(2) Optimize $\boldsymbol{\theta}_{k+1}$ from the following recursion

$$\begin{aligned}\boldsymbol{\theta}_{k+1} &= \boldsymbol{\theta}_k + \omega_{k+1}\left(g_{\boldsymbol{\theta}_k}(\boldsymbol{\beta}_{k+1}) - \boldsymbol{\theta}_k\right) \\ &= (1 - \omega_{k+1})\boldsymbol{\theta}_k + \omega_{k+1}\boldsymbol{g}_{\boldsymbol{\theta}_k}(\boldsymbol{\beta}_{k+1}),\end{aligned} \tag{4}$$

where $\boldsymbol{g}_{\boldsymbol{\theta}_k}(\cdot)$ is some mapping to derive the optimal $\boldsymbol{\theta}$ based on the current $\boldsymbol{\beta}$.

**Remark**: Define $H(\boldsymbol{\theta}_k, \boldsymbol{\beta}_{k+1}) = g_{\boldsymbol{\theta}_k}(\boldsymbol{\beta}_{k+1}) - \boldsymbol{\theta}_k$. In this formulation, our target is to find $\boldsymbol{\theta}^*$ that solves $h(\boldsymbol{\theta}^*) = \mathbb{E}[H(\boldsymbol{\theta}, \boldsymbol{\beta})] = 0$.

**General Assumptions**

To provide the $L_2$ upper bound for SGLD-SA, we first lay out the following assumptions:

**Assumption 1** (Step size and Convexity). *$\{\omega_k\}_{k\in\mathrm{N}}$ is a positive decreasing sequence of real numbers such that*

$$\omega_k \to 0, \ \sum_{k=1}^{\infty} \omega_k = +\infty. \tag{5}$$

*There exist $\delta > 0$ and $\boldsymbol{\theta}^*$ such that for $\boldsymbol{\theta} \in \boldsymbol{\Theta}$:* [†]

$$\langle \boldsymbol{\theta} - \boldsymbol{\theta}^*, h(\boldsymbol{\theta}) \rangle \leq -\delta \|\boldsymbol{\theta} - \boldsymbol{\theta}^*\|^2, \tag{6}$$

*with additionally*

$$\lim_{k\to\infty} \inf 2\delta \frac{\omega_k}{\omega_{k+1}} + \frac{\omega_{k+1} - \omega_k}{\omega_{k+1}{}^2} > 0. \tag{7}$$

*Then for any $\alpha \in (0, 1]$ and suitable $A$ and $B$, a practical $\omega_k$ can be set as*

$$\omega_k = A(k + B)^{-\alpha} \tag{8}$$

**Assumption 2** (Smoothness). *$L(\boldsymbol{\beta}, \boldsymbol{\theta})$ is $M$-smooth with $M > 0$, i.e. for any $\boldsymbol{\beta}, \boldsymbol{\iota} \in \boldsymbol{B}$, $\boldsymbol{\theta}, \boldsymbol{v} \in \boldsymbol{\Theta}$.*

$$\|\nabla_{\boldsymbol{\beta}} L(\boldsymbol{\beta}, \boldsymbol{\theta}) - \nabla_{\boldsymbol{\beta}} L(\boldsymbol{\iota}, \boldsymbol{v})\| \leq M\|\boldsymbol{\beta} - \boldsymbol{\iota}\| + M\|\boldsymbol{\theta} - \boldsymbol{v}\|. \tag{9}$$

**Assumption 3** (Dissipative). *There exist constants $m > 0, b \geq 0$, s.t. for all $\boldsymbol{\beta} \in \boldsymbol{\beta}$ and $\boldsymbol{\theta} \in \boldsymbol{\Theta}$, we have*

$$\langle \nabla_{\boldsymbol{\beta}} L(\boldsymbol{\beta}, \boldsymbol{\theta}), \boldsymbol{\beta} \rangle \leq b - m\|\boldsymbol{\beta}\|^2. \tag{10}$$

**Assumption 4** (Gradient condition). *The stochastic noise $\boldsymbol{\chi}_k \in \boldsymbol{B}$, which comes from $\nabla_{\boldsymbol{\beta}} \tilde{L}(\boldsymbol{\beta}_k, \boldsymbol{\theta}_k) - \nabla_{\boldsymbol{\beta}} L(\boldsymbol{\beta}_k, \boldsymbol{\theta}_k)$, is a white noise or Martingale difference noise and is independent with each other.*

$$\mathbb{E}[\boldsymbol{\chi}_k | \mathcal{F}_k] = 0. \tag{11}$$

*The scale of the noise is bounded by*

$$\mathbb{E}\|\boldsymbol{\chi}\|^2 \leq M^2 \mathbb{E}\|\boldsymbol{\beta}\|^2 + M^2 \mathbb{E}\|\boldsymbol{\theta}\|^2 + B^2. \tag{12}$$

*for constants $M, B > 0$.*

---

[†] $\|\cdot\|$ is short for $\|\cdot\|_2$

In addition to the assumptions, we also assume the existence of Markov transition kernel, the proof goes beyond the scope of our paper.

**Proposition 1.** *There exist constants $M, B > 0$ such that*

$$\|g_{\boldsymbol{\theta}}(\boldsymbol{\beta})\|^2 \le M^2 \|\boldsymbol{\beta}\|^2 + B^2 \tag{13}$$

*Proof.* As shown in Eq.(12), Eq.(13) and Eq.(15) in the main body, $\rho$, $\delta$ and $\kappa$ are clearly bounded. It is also easy to verify that $\sigma$ in Eq.(14) in the main body satisfies (13). For convenience, we choose the same $M$ and $B$ (large enough) as in (12). □

**Proposition 2.** *For any $\boldsymbol{\beta} \in \boldsymbol{B}$, it holds that*

$$\|\nabla_{\boldsymbol{\beta}} L(\boldsymbol{\beta}, \boldsymbol{\theta})\|^2 \le 3M^2 \|\boldsymbol{\beta}\|^2 + 3M^2 \|\boldsymbol{\theta}\|^2 + 3B^2 \tag{14}$$

*for constants $M$ and $B$.*

*Proof.* Suppose there is a minimizer $(\boldsymbol{\theta}^*, \boldsymbol{\beta}^*)$ such that $\nabla_{\boldsymbol{\beta}} L(\boldsymbol{\beta}^*, \boldsymbol{\theta}^*) = 0$ and $\boldsymbol{\theta}^*$ has reached the stationary point, following Assumption 3 we have,

$$\langle \nabla_{\boldsymbol{\beta}} L(\boldsymbol{\beta}^*, \boldsymbol{\theta}^*), \boldsymbol{\beta}^* \rangle \le b - m \|\boldsymbol{\beta}^*\|^2.$$

Therefore, $\|\boldsymbol{\beta}^*\|^2 \le \frac{b}{m}$. Since $\boldsymbol{\theta}^*$ is the stationary point, $\boldsymbol{\theta}^* = (1 - \omega)\boldsymbol{\theta}^* + \omega g_{\boldsymbol{\theta}^*}(\boldsymbol{\beta}^*)$. By (13), we have $\|g_{\boldsymbol{\theta}^*}(\boldsymbol{\beta}^*)\|^2 \le M^2 \|\boldsymbol{\beta}^*\|^2 + B^2$, which implies that $\|\boldsymbol{\theta}^*\|^2 = \|g_{\boldsymbol{\theta}^*}(\boldsymbol{\beta}^*)\|^2 \le M^2 \|\boldsymbol{\beta}^*\|^2 + B^2 \le \frac{b}{m} M^2 + B^2$. By the smoothness assumption 2, we have

$$
\begin{aligned}
&\|\nabla_{\boldsymbol{\beta}} L(\boldsymbol{\beta}, \boldsymbol{\theta})\| \\
\le &\|\nabla_{\boldsymbol{\beta}} L(\boldsymbol{\beta}^*, \boldsymbol{\theta}^*)\| + M \|\boldsymbol{\beta} - \boldsymbol{\beta}^*\| + M \|\boldsymbol{\theta} - \boldsymbol{\theta}^*\| \\
\le &0 + M(\|\boldsymbol{\beta}\| + \sqrt{\frac{b}{m}} + \|\boldsymbol{\theta}\| + \|\boldsymbol{\theta}^*\|) \\
\le &M \|\boldsymbol{\theta}\| + M \|\boldsymbol{\beta}\| + M(\sqrt{\frac{b}{m}} + \sqrt{\frac{b}{m} M^2 + B^2}) \\
\le &M \|\boldsymbol{\theta}\| + M \|\boldsymbol{\beta}\| + \bar{B},
\end{aligned}
$$

where $\bar{B} = M(\sqrt{\frac{b}{m} M^2 + B^2} + \sqrt{\frac{b}{m}})$. Therefore,

$$\|\nabla_{\boldsymbol{\beta}} L(\boldsymbol{\beta}, \boldsymbol{\theta})\|^2 \le 3M^2 \|\boldsymbol{\beta}\|^2 + 3M^2 \|\boldsymbol{\theta}\|^2 + 3\bar{B}^2.$$

For notation simplicity, we can choose the same $B$ (large enough) to bound (12), (13) and (14). □

**Lemma 1** (Uniform $L_2$ bounds). *For all $0 < \epsilon < \text{Re}(\frac{m - \sqrt{m^2 - 4M^2(M^2+1)}}{4M^2(M^2+1)})$, there exist $G, \overline{G} > 0$ such that $\sup \mathbb{E}\|\boldsymbol{\beta}_k\|^2 \le G$ and $\sup \mathbb{E}\|\boldsymbol{\theta}_k\|^2 \le \overline{G}$, where $G = \|\boldsymbol{\beta}_0\|^2 + \frac{1}{m}(b + 2\epsilon B^2(M^2+1) + \tau d)$ and $\overline{G} = M^2 G + B^2$.*

*Proof.* From (3), we have

$$
\begin{aligned}
&\mathbb{E}\|\boldsymbol{\beta}_{k+1}\|^2 \\
=&\mathbb{E}\left\|\boldsymbol{\beta}_k + \epsilon \nabla_{\boldsymbol{\beta}} \tilde{L}(\boldsymbol{\beta}_k, \boldsymbol{\theta}_k)\right\|^2 + 2\tau\epsilon \mathbb{E}\|\boldsymbol{\eta}_k\|^2 + \sqrt{8\epsilon\tau}\mathbb{E}\langle \boldsymbol{\beta}_k + \epsilon \nabla_{\boldsymbol{\beta}} \tilde{L}(\boldsymbol{\beta}_k, \boldsymbol{\theta}_k), \boldsymbol{\eta}_k \rangle \\
=&\mathbb{E}\left\|\boldsymbol{\beta}_k + \epsilon \nabla_{\boldsymbol{\beta}} \tilde{L}(\boldsymbol{\beta}_k, \boldsymbol{\theta}_k)\right\|^2 + 2\tau\epsilon d,
\end{aligned} \tag{15}
$$

Moreover, the first item in (15) can be expanded to

$$
\begin{aligned}
&\mathbb{E}\left\|\boldsymbol{\beta}_k + \epsilon \nabla_{\boldsymbol{\beta}} \tilde{L}(\boldsymbol{\beta}_k, \boldsymbol{\theta}_k)\right\|^2 \\
=&\ \mathbb{E}\|\boldsymbol{\beta}_k + \epsilon \nabla_{\boldsymbol{\beta}} L(\boldsymbol{\beta}_k, \boldsymbol{\theta}_k)\|^2 + \epsilon^2 \mathbb{E}\|\boldsymbol{\chi}_k\|^2 - 2\epsilon \mathbb{E}\left[\mathbb{E}\left(\langle \boldsymbol{\beta}_k + \epsilon \nabla_{\boldsymbol{\beta}} L(\boldsymbol{\beta}_k, \boldsymbol{\theta}_k), \boldsymbol{\chi}_k \rangle | \mathcal{F}_k\right)\right] \\
=&\ \mathbb{E}\|\boldsymbol{\beta}_k + \epsilon \nabla_{\boldsymbol{\beta}} L(\boldsymbol{\beta}_k, \boldsymbol{\theta}_k)\|^2 + \epsilon^2 \mathbb{E}\|\boldsymbol{\chi}_k\|^2,
\end{aligned} \tag{16}
$$

where (11) is used to cancel the inner product item.

Turning to the first item of (16), the dissipivity condition (10) and the boundness of $\nabla_{\boldsymbol{\beta}} L(\boldsymbol{\beta}, \boldsymbol{\theta})$ (14) give us:

$$
\begin{aligned}
&\mathbb{E}\left\|\boldsymbol{\beta}_k + \epsilon \nabla_{\boldsymbol{\beta}} L(\boldsymbol{\beta}_k, \boldsymbol{\theta}_k)\right\|^2 \\
&= \mathbb{E}\|\boldsymbol{\beta}_k\|^2 + 2\epsilon \mathbb{E}\langle \boldsymbol{\beta}_k, \nabla_{\boldsymbol{\beta}} L(\boldsymbol{\beta}_k, \boldsymbol{\theta}_k)\rangle + \epsilon^2 \mathbb{E}\left\|\nabla_{\boldsymbol{\beta}} L(\boldsymbol{\beta}_k, \boldsymbol{\theta}_k)\right\|^2 \\
&\leq \mathbb{E}\|\boldsymbol{\beta}_k\|^2 + 2\epsilon(b - m\mathbb{E}\|\boldsymbol{\beta}_k\|^2) + \epsilon^2(3M^2\mathbb{E}\|\boldsymbol{\beta}_k\|^2 + 3M^2\mathbb{E}\|\boldsymbol{\theta}_k\|^2 + 3B^2) \\
&= (1 - 2\epsilon m + 3\epsilon^2 M^2)\mathbb{E}\|\boldsymbol{\beta}_k\|^2 + 2\epsilon b + 3\epsilon^2 B^2 + 3\epsilon^2 M^2 \mathbb{E}\|\boldsymbol{\theta}_k\|^2.
\end{aligned}
\tag{17}
$$

By (12), the second item of (16) is bounded by

$$
\mathbb{E}\|\boldsymbol{\chi}_k\|^2 \leq M^2 \mathbb{E}\|\boldsymbol{\beta}_k\|^2 + M^2 \mathbb{E}\|\boldsymbol{\theta}_k\|^2 + B^2.
\tag{18}
$$

Combining (15), (16), (17) and (18), we have

$$
\mathbb{E}\|\boldsymbol{\beta}_{k+1}\|^2 \leq (1 - 2\epsilon m + 4\epsilon^2 M^2)\mathbb{E}\|\boldsymbol{\beta}_k\|^2 + 2\epsilon b + 4\epsilon^2 B^2 + 4\epsilon^2 M^2 \mathbb{E}\|\boldsymbol{\theta}_k\|^2 + 2\tau\epsilon d.
\tag{19}
$$

Next we use proof by induction to show for $k = 1, 2, \ldots, \infty$, $\mathbb{E}\|\boldsymbol{\beta}_k\|^2 \leq G$, where

$$
G = \mathbb{E}\|\boldsymbol{\beta}_0\|^2 + \frac{b + 2\epsilon B^2(M^2 + 1) + \tau d}{m - 2\epsilon M^2(M^2 + 1)}.
\tag{20}
$$

First of all, the case of $k = 0, 1$ is trivial. Then if we assume for each $k \in 2, 3, \ldots, t$, $\mathbb{E}\|\boldsymbol{\beta}_k\|^2 \leq G$, $\mathbb{E}\|g(\boldsymbol{\beta}_k)\|^2 \leq M^2 G + B^2$, $\mathbb{E}\|\boldsymbol{\theta}_{k-1}\|^2 \leq M^2 G + B^2$. It follows that,

$$
\begin{aligned}
\mathbb{E}\|\boldsymbol{\theta}_k\|^2 &= \mathbb{E}\|(1 - \omega_k)\boldsymbol{\theta}_{k-1} + \omega_k g(\boldsymbol{\beta}_k)\|^2 \\
&\leq (1 - \omega_k)^2 \mathbb{E}\|\boldsymbol{\theta}_{k-1}\|^2 + \omega_k{}^2 \mathbb{E}\|g(\boldsymbol{\beta}_k)\|^2 + 2(1 - \omega_k)\omega_k \mathbb{E}\langle \boldsymbol{\theta}_{k-1}, g(\boldsymbol{\beta}_k)\rangle \\
&\leq (1 - \omega_k)^2 \mathbb{E}\|\boldsymbol{\theta}_{k-1}\|^2 + \omega_k{}^2 \mathbb{E}\|g(\boldsymbol{\beta}_k)\|^2 + 2(1 - \omega_k)\omega_k \sqrt{\mathbb{E}\|\boldsymbol{\theta}_{k-1}\|^2 \mathbb{E}\|g(\boldsymbol{\beta}_k)\|^2} \\
&\leq (1 - \omega_k)^2 (M^2 G + B^2) + \omega_k{}^2 (M^2 G + B^2) + 2(1 - \omega_k)\omega_k (M^2 G + B^2) \\
&= M^2 G + B^2,
\end{aligned}
$$

Next, we proceed to prove $\mathbb{E}\|\boldsymbol{\beta}_{t+1}\|^2 \leq G$ and $\mathbb{E}\|\boldsymbol{\theta}_{t+1}\|^2 \leq M^2 G + B^2$. Following (19), we have

$$
\begin{aligned}
&\mathbb{E}\|\boldsymbol{\beta}_{t+1}\|^2 \\
&\leq (1 - 2\epsilon m + 4\epsilon^2 M^2)\mathbb{E}\|\boldsymbol{\beta}_k\|^2 + 2\epsilon b + 4\epsilon^2 B + 4\epsilon^2 M^2 \mathbb{E}\|\boldsymbol{\theta}_k\|^2 + 2\tau\epsilon d \\
&\leq (1 - 2\epsilon m + 4\epsilon^2 M^2)G + 2\epsilon b + 4\epsilon^2 B + 4\epsilon^2 M^2(M^2 G + B^2) + 2\tau\epsilon d \\
&\leq \left(1 - 2\epsilon m + 4\epsilon^2 M^2(M^2 + 1)\right)G + 2\epsilon b + 4\epsilon^2 B^2(M^2 + 1) + 2\tau\epsilon d
\end{aligned}
\tag{21}
$$

Consider the quadratic equation $1 - 2mx + 4M^2(M^2 + 1)x^2 = 0$. If $m^2 - 4M^2(M^2 + 1) \geq 0$, then the smaller root is $\frac{m - \sqrt{m^2 - 4M^2(M^2+1)}}{4M^2(M^2+1)}$ which is positive; otherwise the quadratic equation has no real solutions and is always positive. Fix $\epsilon \in \left(0, \mathrm{Re}\left(\frac{m - \sqrt{m^2 - 4M^2(M^2+1)}}{4M^2(M^2+1)}\right)\right)$ so that

$$
0 < 1 - 2\epsilon m + 4\epsilon^2 M^2(M^2 + 1) < 1.
\tag{22}
$$

With (20), we can further bound (21) as follows:

$$
\begin{aligned}
&\mathbb{E}\|\boldsymbol{\beta}_{t+1}\|^2 \\
&\leq \left(1 - 2\epsilon m + 4\epsilon^2 M^2(M^2 + 1)\right)\left(\mathbb{E}\|\boldsymbol{\beta}_0\|^2 + \mathbb{I}\right) + 2\epsilon b + 4\epsilon^2 B^2(M^2 + 1) + 2d\tau\epsilon \\
&= \left(1 - 2\epsilon m + 4\epsilon^2 M^2(M^2 + 1)\right)\mathbb{E}\|\boldsymbol{\beta}_0\|^2 + \mathbb{I} - \left(2\epsilon b + 4\epsilon^2 B^2(M^2 + 1) + 2d\tau\epsilon\right) \\
&\quad + \left(2\epsilon b + 4\epsilon^2 B^2(M^2 + 1) + 2\epsilon\tau d\right) \\
&\leq \mathbb{E}\|\boldsymbol{\beta}_0\|^2 + \mathbb{I} \equiv G,
\end{aligned}
\tag{23}
$$

where $\mathbb{I} = \dfrac{b + 2\epsilon B^2(M^2 + 1) + d\tau}{m - 2\epsilon M^2(M^2 + 1)}$, the second to the last inequality comes from (22).

Moreover, from (13), we also have

$$\mathbb{E}\|g(\boldsymbol{\beta}_{t+1})\|^2 \le M^2\mathbb{E}\|\boldsymbol{\beta}_{t+1}\|^2 + B^2 \le M^2 G + B^2,$$

$$
\begin{aligned}
\mathbb{E}\|\boldsymbol{\theta}_{t+1}\|^2 &= \mathbb{E}\|(1 - \omega_{t+1})\boldsymbol{\theta}_t + \omega_{t+1}g(\boldsymbol{\beta}_{t+1})\|^2 \\
&\le (1 - \omega_{t+1})^2\mathbb{E}\|\boldsymbol{\theta}_t\|^2 + \omega_{t+1}^2\mathbb{E}\|g(\boldsymbol{\beta}_{t+1})\|^2 + 2(1 - \omega_{t+1})\omega_{t+1}\mathbb{E}\langle\boldsymbol{\theta}_t, g(\boldsymbol{\beta}_{t+1})\rangle \\
&\le (1 - \omega_{t+1})^2\mathbb{E}\|\boldsymbol{\theta}_t\|^2 + \omega_{t+1}^2\mathbb{E}\|g(\boldsymbol{\beta}_{t+1})\|^2 + 2(1 - \omega_{t+1})\omega_{t+1}\sqrt{\mathbb{E}\|\boldsymbol{\theta}_t\|^2\mathbb{E}\|g(\boldsymbol{\beta}_{t+1})\|^2} \\
&\le (1 - \omega_{t+1})^2(M^2 G + B^2) + \omega_{t+1}^2(M^2 G + B^2) + 2(1 - \omega_{t+1})\omega_{t+1}(M^2 G + B^2) \\
&= M^2 G + B^2,
\end{aligned}
$$

Therefore, we have proved that for any $k \in 1, 2, \ldots, \infty$, $\mathbb{E}\|\boldsymbol{\beta}_k\|^2$, $\mathbb{E}\|g(\boldsymbol{\beta}_k)\|^2$ and $\mathbb{E}\|\boldsymbol{\theta}_k\|^2$ are bounded. Furthermore, we notice that $G$ can be unified to a constant $G = \mathbb{E}\|\boldsymbol{\beta}_0\|^2 + \frac{1}{m}\left(b + 2\epsilon B^2(M^2 + 1) + \tau d\right)$. $\square$

**Assumption 5** (Solution of Poisson equation). *For all $\boldsymbol{\theta} \in \boldsymbol{\Theta}$, there exists a function $\mu_{\boldsymbol{\theta}}$ on $\boldsymbol{\beta}$ that solves the Poisson equation $\mu_{\boldsymbol{\theta}}(\boldsymbol{\beta}) - \Pi_{\boldsymbol{\theta}}\mu_{\boldsymbol{\theta}}(\boldsymbol{\beta}) = H(\boldsymbol{\theta}, \boldsymbol{\beta}) - h(\boldsymbol{\theta})$, which follows that*

$$H(\boldsymbol{\theta}_k, \boldsymbol{\beta}_{k+1}) = h(\boldsymbol{\theta}_k) + \mu_{\boldsymbol{\theta}_k}(\boldsymbol{\beta}_{k+1}) - \Pi_{\boldsymbol{\theta}_k}\mu_{\boldsymbol{\theta}_k}(\boldsymbol{\beta}_{k+1}). \tag{24}$$

*There exists a constant $C$ such that for all $\boldsymbol{\theta} \in \boldsymbol{\Theta}$, $\Pi_{\boldsymbol{\theta}}\mu$ is bounded, i.e.*

$$\|\Pi_{\boldsymbol{\theta}}\mu_{\boldsymbol{\theta}}\| \le C \tag{25}$$

We leave the relaxation of the above assumption for future work.

**Proposition 3.** *There exists a constant $C_1$ so that*

$$\mathbb{E}_{\boldsymbol{\theta}}[\|H(\boldsymbol{\theta}, \boldsymbol{\beta})\|^2] \le C_1(1 + \|\boldsymbol{\theta} - \boldsymbol{\theta}^*\|^2) \tag{26}$$

*Proof.* By (13), we have

$$\mathbb{E}\|g_{\boldsymbol{\theta}}(\boldsymbol{\beta}) - \boldsymbol{\theta}\|^2 \le 2\mathbb{E}\|g_{\boldsymbol{\theta}}(\boldsymbol{\beta})\|^2 + 2\|\boldsymbol{\theta}\|^2 \le 2(M^2\mathbb{E}\|\boldsymbol{\beta}\|^2 + B^2) + 2\|\boldsymbol{\theta}\|^2$$

Since we have proved the $L_2$ boundedness of $\mathbb{E}\|\boldsymbol{\beta}\|^2$, choose $C' = \max(2, 2(M^2\mathbb{E}\|\boldsymbol{\beta}\|^2 + B^2))$, we have

$$\mathbb{E}_{\boldsymbol{\theta}}[\|H(\boldsymbol{\theta}, \boldsymbol{\beta})\|^2] \le C'(1 + \|\boldsymbol{\theta}\|^2) = C'(1 + \|\boldsymbol{\theta} - \boldsymbol{\theta}^* + \boldsymbol{\theta}^*\|^2) \le C_1(1 + \|\boldsymbol{\theta} - \boldsymbol{\theta}^*\|^2)$$

$\square$

Lemma 2 is a restatement of Lemma 25 (page 247) from Benveniste et al. [1990].

**Lemma 2.** *Suppose $k_0$ is an integer which satisfies with*

$$\inf_{k \ge k_0} \frac{\omega_{k+1} - \omega_k}{\omega_k \omega_{k+1}} + 2\delta - \omega_{k+1}C_1 > 0.$$

*Then for any $k > k_0$, the sequence $\{\Lambda_k^K\}_{k=k_0,\ldots,K}$ defined below is increasing*

$$
\begin{cases}
2\omega_k \prod_{j=k}^{K-1}(1 - 2\omega_{j+1}\delta + \omega_{j+1}^2 C_1) & \text{if } k < K, \\
2\omega_k & \text{if } k = K.
\end{cases} \tag{27}
$$

**Lemma 3.** *There exist $\lambda_0$ and $k_0$ such that for all $\lambda \ge \lambda_0$ and $k \ge k_0$, the sequence $u_k = \lambda\omega_k$ satisfies*

$$u_{k+1} \ge (1 - 2\omega_{k+1}\delta + \omega_{k+1}{}^2 C_1)u_k + \omega_{k+1}{}^2 C_1 + \omega_{k+1}\overline{C}_1. \tag{28}$$

*Proof.* Replace $u_k = \lambda\omega_k$ in (28), we have

$$\lambda\omega_{k+1} \geq (1 - 2\omega_{k+1}\delta + \omega_{k+1}{}^2 C_1)\lambda\omega_k + \omega_{k+1}{}^2 C_1 + \omega_{k+1}\overline{C}_1. \tag{29}$$

According to (7) in assumption 1, we denote $\lim_{k\to\infty}\inf 2\delta\omega_{k+1}\omega_k + \omega_{k+1} - \omega_k$ by $\Delta_+$. Then the above inequality (29) can be simplified as

$$\lambda(\Delta_+ - \omega_{k+1}{}^2\omega_k C_1) \geq \omega_{k+1}{}^2 C_1 + \omega_{k+1}\overline{C}_1. \tag{30}$$

Since the LHS increases to $\Delta_+$ and the RHS decreases to 0 as $k \to \infty$. There exist $\lambda_0$ and $k_0$ such that for all $\lambda > \lambda_0$ and $k > k_0$, (30) holds. $\square$

**Theorem 1** ($L_2$ convergence rate). *Suppose that Assumptions 1-5 hold, there exists a constant $\lambda$ such that*

$$\mathbb{E}\left[\|\boldsymbol{\theta}_k - \boldsymbol{\theta}^*\|^2\right] \leq \lambda\omega_k,$$

*Proof.* Denote $\boldsymbol{T}_k = \boldsymbol{\theta}_k - \boldsymbol{\theta}^*$, with the help of (4) and Poisson equation (24), we deduce that

$$
\begin{aligned}
&\|\boldsymbol{T}_{k+1}\|^2\\
=&\|\boldsymbol{T}_k\|^2 + \omega_{k+1}{}^2\|H(\boldsymbol{\theta}_k, \boldsymbol{\beta}_{k+1})\|^2 + 2\omega_{k+1}\langle\boldsymbol{T}_k, H(\boldsymbol{\theta}_k, \boldsymbol{\beta}_{k+1})\rangle\\
=&\|\boldsymbol{T}_k\|^2 + \omega_{k+1}{}^2\|H(\boldsymbol{\theta}_k, \boldsymbol{\beta}_{k+1})\|^2 + 2\omega_{k+1}\langle\boldsymbol{T}_k, h(\boldsymbol{\theta}_k)\rangle + 2\omega_{k+1}\langle\boldsymbol{T}_k, \mu_{\boldsymbol{\theta}_k}(\boldsymbol{\beta}_{k+1}) - \Pi_{\boldsymbol{\theta}_k}\mu_{\boldsymbol{\theta}_k}(\boldsymbol{\beta}_{k+1})\rangle\\
=&\|\boldsymbol{T}_k\|^2 + \text{D1} + \text{D2} + \text{D3}.
\end{aligned}
$$

First of all, according to (26) and (6), we have

$$\omega_{k+1}{}^2\|H(\boldsymbol{\theta}_k, \boldsymbol{\beta}_{k+1})\|^2 \leq \omega_{k+1}{}^2 C_1(1 + \|\boldsymbol{T}_k\|^2), \tag{D1}$$

$$2\omega_{k+1}\langle\boldsymbol{T}_k, h(\boldsymbol{\theta}_k)\rangle \leq -2\omega_{k+1}\delta\|\boldsymbol{T}_k\|^2, \tag{D2}$$

Conduct the decomposition of D3 similar to Theorem 24 (p.g. 246) from Benveniste et al. [1990] and Lemma A.5 [Liang, 2010].

$$
\begin{aligned}
&\mu_{\boldsymbol{\theta}_k}(\boldsymbol{\beta}_{k+1}) - \Pi_{\boldsymbol{\theta}_k}\mu_{\boldsymbol{\theta}_k}(\boldsymbol{\beta}_{k+1})\\
=&\underbrace{\mu_{\boldsymbol{\theta}_k}(\boldsymbol{\beta}_{k+1}) - \Pi_{\boldsymbol{\theta}_k}\mu_{\boldsymbol{\theta}_k}(\boldsymbol{\beta}_k)}_{\text{D3-1}} + \underbrace{\Pi_{\boldsymbol{\theta}_k}\mu_{\boldsymbol{\theta}_k}(\boldsymbol{\beta}_k) - \Pi_{\boldsymbol{\theta}_{k-1}}\mu_{\boldsymbol{\theta}_{k-1}}(\boldsymbol{\beta}_k)}_{\text{D3-2}} + \underbrace{\Pi_{\boldsymbol{\theta}_{k-1}}\mu_{\boldsymbol{\theta}_{k-1}}(\boldsymbol{\beta}_k) - \Pi_{\boldsymbol{\theta}_k}\mu_{\boldsymbol{\theta}_k}(\boldsymbol{\beta}_{k+1})}_{\text{D3-3}}.
\end{aligned}
$$

(i) $\mu_{\boldsymbol{\theta}_k}(\boldsymbol{\beta}_{k+1}) - \Pi_{\boldsymbol{\theta}_k}\mu_{\boldsymbol{\theta}_k}(\boldsymbol{\beta}_k)$ forms a martingale difference sequence such that

$$\mathbb{E}\left[\mu_{\boldsymbol{\theta}_k}(\boldsymbol{\beta}_{k+1}) - \Pi_{\boldsymbol{\theta}_k}\mu_{\boldsymbol{\theta}_k}(\boldsymbol{\beta}_k)|\mathcal{F}_k\right] = 0. \tag{D3-1}$$

(ii) From Lemma 1, we have that $\mathbb{E}[\|\boldsymbol{T}_k\|]$ is bounded. $\|\Pi_{\boldsymbol{\theta}_k}\mu_{\boldsymbol{\theta}_k}\|$ is also bounded according to (25). Therefore, together with Cauchy–Schwarz inequality, there exists a positive constant $C_2$ such that

$$\mathbb{E}\left[2\omega_{k+1}\langle\boldsymbol{T}_k, \Pi_{\boldsymbol{\theta}_k}\mu_{\boldsymbol{\theta}_k}(\boldsymbol{\beta}_k) - \Pi_{\boldsymbol{\theta}_{k-1}}\mu_{\boldsymbol{\theta}_{k-1}}(\boldsymbol{\beta}_k)\rangle\right] \leq \omega_{k+1}C_2. \tag{D3-2}$$

(iii) D3-3 can be further decomposed to D3-3a and D3-3b

$$
\begin{aligned}
&\langle\boldsymbol{T}_k, \Pi_{\boldsymbol{\theta}_{k-1}}\mu_{\boldsymbol{\theta}_{k-1}}(\boldsymbol{\beta}_k) - \Pi_{\boldsymbol{\theta}_k}\mu_{\boldsymbol{\theta}_k}(\boldsymbol{\beta}_{k+1})\rangle\\
=&\left(\langle\boldsymbol{T}_k, \Pi_{\boldsymbol{\theta}_{k-1}}\mu_{\boldsymbol{\theta}_{k-1}}(\boldsymbol{\beta}_k)\rangle - \langle\boldsymbol{T}_{k+1}, \Pi_{\boldsymbol{\theta}_k}\mu_{\boldsymbol{\theta}_k}(\boldsymbol{\beta}_{k+1})\rangle\right) + \left(\langle\boldsymbol{T}_{k+1}, \Pi_{\boldsymbol{\theta}_k}\mu_{\boldsymbol{\theta}_k}(\boldsymbol{\beta}_{k+1})\rangle - \langle\boldsymbol{T}_k, \Pi_{\boldsymbol{\theta}_k}\mu_{\boldsymbol{\theta}_k}(\boldsymbol{\beta}_{k+1})\rangle\right)\\
=&\underbrace{(\boldsymbol{z}_k - \boldsymbol{z}_{k+1})}_{\text{D3-3a}} + \underbrace{\langle\boldsymbol{T}_{k+1} - \boldsymbol{T}_k, \Pi_{\boldsymbol{\theta}_k}\mu_{\boldsymbol{\theta}_k}(\boldsymbol{\beta}_{k+1})\rangle}_{\text{D3-3b}}.
\end{aligned}
$$

where $\boldsymbol{z}_k = \langle\boldsymbol{T}_k, \Pi_{\boldsymbol{\theta}_{k-1}}\mu_{\boldsymbol{\theta}_{k-1}}(\boldsymbol{\beta}_k)\rangle$. Similar to (ii), there exists a constant $C_3$ such that

$$\mathbb{E}\left[2\omega_{k+1}\langle\boldsymbol{T}_{k+1} - \boldsymbol{T}_k, \Pi_{\boldsymbol{\theta}_k}\mu_{\boldsymbol{\theta}_k}(\boldsymbol{\beta}_{k+1})\rangle\right] \leq C_3\omega_{k+1}$$

Finally, add all the items D1, D2 and D3 together, for some $\overline{C}_1 = C_2 + C_3$, we have

$$\mathbb{E}\left[\|\boldsymbol{T}_{k+1}\|^2\right] \leq (1 - 2\omega_{k+1}\delta + \omega_{k+1}{}^2 C_1)\mathbb{E}\left[\|\boldsymbol{T}_k\|^2\right] + \omega_{k+1}{}^2 C_1 + \omega_{k+1}\overline{C}_1 + 2\omega_{k+1}\mathbb{E}[z_k - z_{k+1}].$$

Moreover, from (25), there exists a constant $C_4$ such that

$$\mathbb{E}[|\boldsymbol{z}_k|] \leq C_4. \tag{31}$$

Lemma 4 is an extension of Lemma 26 (page 248) from Benveniste et al. [1990].

**Lemma 4.** *Let $\{u_k\}_{k \geq k_0}$ as a sequence of real numbers such that for all $k \geq k_0$, some suitable constants $\overline{C}_1$ and $C_1$*

$$u_{k+1} \geq u_k \left(1 - 2\omega_{k+1}\delta + \omega_{k+1}{}^2 C_1\right) + \omega_{k+1}{}^2 C_1 + \omega_{k+1}\overline{C}_1, \tag{32}$$

*and assume there exists such $k_0$ that*

$$\mathbb{E}\left[\|\boldsymbol{T}^{(k_0)}\|^2\right] \leq u^{(k_0)}. \tag{33}$$

*Then for all $k > k_0$, we have*

$$\mathbb{E}\left[\|\boldsymbol{T}_k\|^2\right] \leq u_k + \sum_{j=k_0+1}^{k} \Lambda_j^k (\boldsymbol{z}^{(j-1)} - \boldsymbol{z}^{(j)}).$$

**Proof of Theorem 1 (Continued).** From Lemma 3, we can choose $\lambda_0$ and $k_0$ which satisfy the conditions (32) and (33)

$$\mathbb{E}[\|\boldsymbol{T}^{(k_0)}\|^2] \leq u^{(k_0)} = \lambda_0 \omega^{(k_0)}.$$

From Lemma 4, it follows that for all $k > k_0$

$$\mathbb{E}\left[\|\boldsymbol{T}_k\|^2\right] \leq u_k + \mathbb{E}\left[\sum_{j=k_0+1}^{k} \Lambda_j^k \left(\boldsymbol{z}^{(j-1)} - \boldsymbol{z}^{(j)}\right)\right]. \tag{34}$$

From (31) and the increasing property of $\Lambda_j^k$ in Lemma 2, we have

$$
\begin{aligned}
&\mathbb{E}\left[\left\|\sum_{j=k_0+1}^{k} \Lambda_j^k \left(\boldsymbol{z}^{(j-1)} - \boldsymbol{z}^{(j)}\right)\right\|\right] \\
=&\mathbb{E}\left[\left\|\sum_{j=k_0+1}^{k-1} (\Lambda_{j+1}^k - \Lambda_j^k)\boldsymbol{z}^{(j)} - 2\omega_k \boldsymbol{z}_k + \Lambda_{k_0+1}^k \boldsymbol{z}^{(k_0)}\right\|\right] \\
\leq& \sum_{j=k_0+1}^{k-1} (\Lambda_{j+1}^k - \Lambda_j^k)C_4 + \mathbb{E}[|2\omega_k \boldsymbol{z}_k|] + \Lambda_k^k C_4 \\
\leq&(\Lambda_k^k - \Lambda_{k_0}^k)C_4 + \Lambda_k^k C_4 + \Lambda_k^k C_4 \\
\leq&3\Lambda_k^k C_4 = 6C_4\omega_k.
\end{aligned}
\tag{35}
$$

Therefore, given the sequence $u_k = \lambda_0 \omega_k$ that satisfies conditions (32), (33) and Lemma 4, for any $k > k_0$, from (34) and (35), we have

$$\mathbb{E}[\|\boldsymbol{T}_k\|^2] \leq u_k + 3C_4\Lambda_k^k = (\lambda_0 + 6C_4)\,\omega_k = \lambda\omega_k,$$

where $\lambda = \lambda_0 + 6C_4$. $\qquad\square$

Table 1: Predictive errors in logistic regression based on a test set considering different $v_0$ and $\sigma$

| MAE / MSE | $v_0$=0.01, $\sigma$=1 | $v_0$=0.001, $\sigma$=1 | $v_0$=0.01, $\sigma$=2 | $v_0$=0.001, $\sigma$=2 |
|---|---|---|---|---|
| SGLD-SA | **0.177 / 0.108** | **0.188 / 0.114** | **0.182 / 0.116** | **0.187 / 0.113** |
| SGLD-EM | 0.207 / 0.131 | 0.361 / 0.346 | 0.204 / 0.132 | 0.376 / 0.360 |
| SGLD | 0.295 / 0.272 | 0.335 / 0.301 | 0.350 / 0.338 | 0.337 / 0.319 |

## 2.2 Weak Convergence of Samples

In statistical models with latent variables, the gradient is often biased due to the use of stochastic approximation. Langevin Monte Carlo with inaccurate gradients has been studied by Chen et al. [2015], Dalalyan and Karagulyan [2018], which are helpful to prove the weak convergence of samples. Following theorem 2 in Chen et al. [2015], we have

**Corollary 1.** *Under assumptions in Appendix B.1 and the assumption 1 (smoothness and boundness on the solution functional) in Chen et al. [2015], the distribution of $\boldsymbol{\beta}_k$ converges weakly to the target posterior as $\epsilon \to 0$ and $k \to \infty$.*

*Proof.* Since $\boldsymbol{\theta}_k$ converges to $\boldsymbol{\theta}^*$ in SGLD-SA under assumptions in Appendix B.1 and the gradient is M-smooth (9), we decompose the stochastic gradient $\nabla_{\boldsymbol{\beta}}\tilde{L}(\boldsymbol{\beta}_k, \boldsymbol{\theta}_k)$ as $\nabla_{\boldsymbol{\beta}}L(\boldsymbol{\beta}_k, \boldsymbol{\theta}^*) + \boldsymbol{\xi}_k + \mathcal{O}(k^{-\alpha})$, where $\nabla_{\boldsymbol{\beta}}L(\boldsymbol{\beta}_k, \boldsymbol{\theta}^*)$ is the exact gradient, $\boldsymbol{\xi}_k$ is a zero-mean random vector, $\mathcal{O}(k^{-\alpha})$ is the bias term coming from the stochastic approximation and $\alpha \in (0, 1]$ is used to guarantee the consistency in theorem 1. Therefore, Eq.(3) can be written as

$$\boldsymbol{\beta}_{k+1} = \boldsymbol{\beta}_k + \epsilon_k \left( \nabla_{\boldsymbol{\beta}}L(\boldsymbol{\beta}_k, \boldsymbol{\theta}^*) + \boldsymbol{\xi}_k + \mathcal{O}(k^{-\alpha}) \right) + \sqrt{2\epsilon_k}\boldsymbol{\eta}_k, \text{ where } \boldsymbol{\eta}_k \sim \mathcal{N}(0, \boldsymbol{I}). \quad (36)$$

Following a similar proof in Chen et al. [2015], it suffices to show that $\sum_{k=1}^{K} k^{-\alpha}/K \to 0$ as $K \to \infty$, which is obvious. Therefore, the distribution of $\boldsymbol{\beta}_k$ converges weakly to the target distribution as $\epsilon \to 0$ and $k \to \infty$. □

## 3 Simulation of Large-p-Small-n Logistic Regression

Now we conduct the experiments on binary logistic regression. The setup is similar as before, except $n$ is set to 500, $\Sigma_{i,j} = 0.3^{|i-j|}$ and $\boldsymbol{\eta} \sim \mathcal{N}(0, \boldsymbol{I}/2)$. We set the learning rate for all the three algorithms to $0.001 \times k^{-\frac{1}{3}}$ and step size $\omega_k$ to $10 \times (k + 1000)^{-0.7}$. The binary response values are simulated from **Bernoulli**$(p)$ where $p = 1/(1 + e^{-X\boldsymbol{\beta} - \boldsymbol{\eta}})$. As shown in Fig.1: SGLD fails in selecting the right variables and overfits the data; both SGLD-EM and SGLD-SA choose the right variables. However, SGLD-EM converges to a poor local optimum by mistakenly using $L_1$ norm to regularize all the variables, leading to a large shrinkage effect on $\boldsymbol{\beta}_{1:3}$. By contrast, SGLD-SA successfully updates the latent variables and regularize $\beta_{1:3}$ with $L_2$ norm, yielding a better parameter estimation for $\beta_{1:3}$ and a stronger regularization for $\beta_{4-1000}$. Table.1 illustrates that SGLD-SA consistently outperforms the other methods and is robust to different initializations. We observe that SGLD-EM sometimes performs as worse as SGLD when $v_0 = 0.001$, which indicates that the EM-based variable selection is not robust in the stochastic optimization of the latent variables.

## 4 Regression on UCI datasets

We further evaluate our model on five **UCI** regression datasets and show the results in Table 2. Following Hernandez-Lobato and Adams [2015], we randomly sample 90% of each dataset for training and leave the rest for testing. We run 20 experiments for each setup with fixed random seeds and report the averaged error rate. Feature normalization is applied in the experiments. The model is a simple MLP with one hidden layer of 50 units. We set the batch size to 50, the training epoch to 200, the learning rate to 1e-5 and the default $L_2$ to 1e-4. For SGHMC-EM and SGHMC-SA, we apply the SSGL prior on the BNN weights (excluding biases) and fix $a, \nu, \lambda = 1, b, v_1, \sigma = 10$ and $\delta = 0.5$. We fine-tune the initial temperature $\tau$ and $v_0$. As shown in Table 2, SGHMC-SA outperforms all the baselines. Nevertheless, without smooth adaptive update, SGHMC-EM often performs worse than SGHMC. While with simulated annealing where $\tau^{(k)} = \tau \times 1.003^k$, we observe further improved performance in most of the cases.

Figure 1: Logistic regression simulation when $v_0 = 0.1$ and $\sigma = 1$

| Dataset | Boston | Yacht | Energy | Wine | Concrete |
|---------|--------|-------|--------|------|----------|
| Hyperparameters | 1/0.1 | 1/0.1 | 0.1/0.1 | 0.5/0.01 | 0.5/0.07 |
| SGHMC | 2.783±0.109 | 0.886±0.046 | 1.983±0.092 | 0.731±0.015 | 6.319±0.179 |
| A-SGHMC | 2.848±0.126 | 0.808±0.048 | 1.419±0.067 | 0.671±0.019 | 5.978±0.166 |
| SGHMC-EM | 2.813±0.140 | 0.823±0.053 | 2.077±0.108 | 0.729±0.018 | 6.275±0.169 |
| A-SGHMC-EM | 2.767±0.154 | 0.815±0.052 | 1.435±0.069 | 0.627±0.008 | 5.762±0.156 |
| SGHMC-SA | **2.779±0.133** | **0.789±0.050** | **1.948±0.081** | **0.654±0.010** | **6.029±0.131** |
| A-SGHMC-SA | **2.692±0.120** | **0.782±0.052** | **1.388±0.052** | **0.620±0.008** | **5.687±0.142** |

Table 2: Average performance and standard deviation of Root Mean Square Error, where $\tau$ denotes the initial inverse temperature and $v_0$ is a hyperparameter in the SSGL prior (Hyperparameters $\tau/v_0$).

# 5 Experimental Setup

## 5.1 Network Architecture

The first DNN we use is a standard 2-Conv-2-FC CNN: it has two convolutional layers with a $2 \times 2$ max pooling after each layer and two fully-connected layers. The filter size in the convolutional layers is $5 \times 5$ and the feature maps are set to be 32 and 64, respectively [Jarrett et al., 2009]. The fully-connected layers (FC) have 200 hidden nodes and 10 outputs. We use the rectified linear unit (ReLU) as activation function between layers and employ a cross-entropy loss.

The second DNN is a 2-Conv-BN-3-FC CNN: it has two convolutional layers with a $2 \times 2$ max pooling after each layer and three fully-connected layers with batch normalization applied to the first FC layer. The filter size in the convolutional layers is $4 \times 4$ and the feature maps are both set to 64. We use $256 \times 64 \times 10$ fully-connected layers.

## 5.2 Data Augmentation

The MNIST dataset is augmented by (1) randomCrop: randomly crop each image with size 28 and padding 4, (2) random rotation: randomly rotate each image by a degree in $[-15°, +15°]$, (3) normalization: normalize each image with empirical mean 0.1307 and standard deviation 0.3081.

The FMNIST dataset is augmented by (1) randomCrop: same as MNIST, (2) randomHorizontalFlip: randomly flip each image horizontally, (3) normalization: same as MNIST, (4) random erasing [Zhong et al., 2017].

The CIFAR10 dataset is augmented by (1) randomCrop: randomly crop each image with size 32 and padding 4, (2) randomHorizontalFlip: randomly flip each image horizontally, (3) normalization: normalize each image with empirical mean (0.4914, 0.4822, 0.4465) and standard deviation (0.2023, 0.1994, 0.2010), (4) random erasing.