[Reviews · NeurIPS 2019]

Reviewer 1



This is a novel combination of existing techniques that appears well-formulated with intriguing experimental results. In particular, this work leverages the strengths stochastic gradient MCMC methods with stochastic approximation to form an adaptive empirical Bayesian approach to learning the parameters and hyperparameters of a Bayesian neural network (BNN). My best understanding is that by optimizing the hyperparameters (rather than sampling them), this new method improves upon existing approaches, speeding up inference without sacrificing quality (especially in the model compression domain). Other areas of BNN literature could be cited, but I think the authors were prudent not to distract the reader from the particular area of focus. This work demonstrates considerable theoretical analysis and is supported by intriguing experimental evidence. In particular, applying the method to a simpler problem (Bayesian linear regression with p << n) shows posteriors matching the data generating distributions, appropriately shrinks irrelevant variables, and attains better predictive performance than similar methods without the contributions of this paper. Similar predictive performance seems to hold for MNIST and FMNIST compared to SGHMC (considered by some to be a “gold standard” in learning BNN parameters), and in a compression setting on CIFAR10. My only complaint about the experimental section would be wanting to see more datasets (perhaps even some regression datasets other than the toy example [that was by no means trivial]), but I think the paper is sufficient as-is. The authors go to considerable lengths to demonstrate the theory of their contribution without distracting the reader from the main results, appropriately deferring lengthy proofs to supplemental material. I think that, coupled with the supplemental material, an expert would be able to reproduce the results in this paper, especially someone already familiar with SGLD/SGHMC methods. Sections 3.2-3.3 are a bit dense to read through, but considering the complexity of the approach and the desire for work to be reproducible, I think this is appropriate. I think the results on compression in MNIST and the uncertainty in the posterior for the toy example are important. They show the strength of this method (even in a neural network setting) and its ability to prune extraneous model parameters, which has positive implications for embedded devices and other lower-memory applications. I think future researchers could indeed apply this method as-is to certain models, although some hyperparameter guidance (e.g., \tau, the temperature) would be nice for practitioners. Especially because of the theoretical guarantees of the method, I think the results make it a more compelling alternative than SGLD/SGHMC for Bayesian inference, especially where sparsity is concerned. **post-author feedback** I appreciate the authors' response, especially regarding having some additional results on UCI regression datasets, showing an improvement over SGHMC (in particular). And I appreciate the authors' response to the other reviewers with regard to a clear problem statement and possible extensions of the work (larger networks, more structure to the sparsity).

Reviewer 2



The paper proposes a novel adaptive empirical Bayesian method to train sparse Bayesian neural networks. The proposed method works by alternately sampling the network parameters from the posterior distribution using stochastic gradient Markov Chain Monte Carlo (MCMC) and smoothly optimizing the hyperparameters of the prior distribution using stochastic approximation (SA). Originality: the proposed sampling scheme enables learning BNNs of complex forms and seems novel. However, I am still unclear as to exactly what limitations of the previous related methods that this work aimed to address, and what was the key ingredient to enabling such advance. Quality: I believe the work is technically sound and the model assumptions are clearly stated. However, the authors do not discuss the weaknesses of the method. Are there any caveats to practitioners due to some violation of the assumptions given in Appendix. B or for any other reasons? Clarity: the writing is highly technical and rather dense, which I understand is necessary for some parts. However, I believe the manuscript would be readable to a broader audience if Sections 2 and 3 are augmented with more intuitive explanations of the motivations and their proposed methods. Many details of the derivations could be moved to the appendix and the resultant space could be used to highlight the key machinery which enabled efficient inference and to develop intuitions. Many terms and notations are not defined in text (as raised in "other comments" below). Significance: the empirical results support the practical utility of the method. I am not sure, however, if the experiments on synthetic datasets, support the theoretical insights presented in the paper. I believe that the method is quite complex and recommend that the authors release the codes to maximize the impact. Other comments: - line 47 - 48 "over-parametrization invariably overfits the data and results in worse performance": over-parameterization seems to be very helpful for supervised learning of deep neural networks in practice ... Also, I have seen a number of theoretical work showing the benefits of over-parametrisation e.g. [1]. - line 71: $\beta$ is never defined. It denotes the set of model parameters, right? - line 149-150 "the convergence to the asymptotically correct distribution allows ... obtain better point estimates in non-convex optimization.": this is only true if the assumptions in Appendix. B are satisfied, isn't it? How realistic are these assumptions in practice? - line 1: MCMC is never defined: Markov Chain Monte Carlo - line 77: typo "gxc lobal"=> "global" - eq.4: $\mathcal{N}$ and $\mathcal{L}$ are not defined. Normal and Laplace I suppose. You need to define them, please. - Table 2: using the letter a to denote the difference in used models is confusing. - too many acronyms are used. References: [1] Allen-Zhu, Zeyuan, Yuanzhi Li, and Zhao Song. "A convergence theory for deep learning via over-parameterization." arXiv preprint arXiv:1811.03962 (2018). ---------------------------------------------------------------------- I am grateful that the authors have addressed most of the concerns about the paper, and have updated my score accordingly. I would like to recommend for acceptance provided that the authors reflect the given clarifications in the paper.

Reviewer 3



This paper combines spike-slab prior, SGMCMC and stochastic approximation to prune the structure of neural networks. The author proposes to use SA to optimize the meta parameters, such as spike-slab selection parameter gamma, and use SGMCMC to get the posterior distribution of weights. Right now the pruning seems to be done in a per scalar fashion. It would be morenteresting if the authors could study more structural version of pruning. E.g. use spike-slab to select which pathway to turn on and off, so we can have more structured sparsity pattern that can be speedup more easily. Most of the current experimental study are focused on small neural networks, what would it take to scale the experimental results to bigger datasets and models? We could also argue that SG-MCMC-SA works better for small neural network domain. Some discussions on this would be helpful.

[Author Response · NeurIPS 2019]

We thank all the reviewers for the valuable comments and suggestions.

To Reviewer 1

#1. Regression experiments on UCI regression datasets.

We further evaluate our model on five **UCI** regression datasets and show the results in Table 1. We randomly sample
90% of each dataset for training and leave the rest for testing. We run 20 experiments for each setup with fixed random
seeds and report the averaged error rate. Feature normalization is applied in the experiments. The model is a simple
MLP with one hidden layer of 50 units. We set the batch size to 50, the training epoch to 200, the learning rate to 1e-4,
the default $L^2$ to 0.003 and the initial inverse temperature $\tau$ to 300. For SGHMC-EM and SGHMC-SA, we apply the
SSGL prior on the BNN weights (excluding biases) and fix $a, \nu, \lambda = 1$, $v_1, \sigma = 10$ and $\delta = 0.5$. We select $b$ from
$\{10, 100\}$, $v_0$ from $\{0.001, 0.01, 0.1\}$. As shown in Table 1, SGHMC-SA outperforms all baselines. Nevertheless,
without smooth adaptive update, SGHMC-EM mostly performs worse than SGHMC. While with simulated annealing
where $\tau^{(k)} = 300 \times r^k$, we observe further improved performance in most of the cases with the optimal rate $r$ selected
from $\{1.01, 1.015, 1.02\}$. We plan to include the distributional distance metrics and other results in the future revision.

To Reviewer 2

# 1. Writing suggestions.

We appreciate the suggestions on writing and are to fix them in the future revision.

# 2. Problem statement and solution.

This paper provides a systematic approach for conducting sparse deep learning with two innovations: (i) We propose to
use the spike-and-slab prior to shrink and cluster the connection weights to two clusters, which facilitates the followed
weight pruning procedure; (ii) We propose an adaptive SGMCMC algorithm to automatically tune the hyper-parameters
of the spike-and-slab prior and prove the convergence of the SGMCMC algorithm rigorously. The adaptive SGMCMC
algorithm is itself of interest, which can be used in many "big data" applications, for example, estimating parameters
for a state-space model when the states are simulated using a SGMCMC algorithm.

# 3. Over-parameterization and how realistic are these assumptions.

We acknowledge over-parameterization may fit some real applications better under certain scenarios. Our assumptions
are quite standard in the adaptive sampling literatures and we have already made efforts to loose the assumptions, such
as Lemma 1 in the appendix. We leave the extension on weaker assumptions in the future.

To Reviewer 3

# 1. Use spike-and-slab to select the structure.

Thanks for the constructive comments. We include scalar-fashion pruning to strengthen the predictive power as Resnet
is a complicated model. We run additional experiments on **UCI** datasets with standard BNNs, and observe iterative
pruning based on suitable probability thresholds can obtain good performance. E.g., on the **Wine** dataset, when pruned
with $\rho$ lower than 0.3, the model ends up with 31% sparsity in the hidden layer and 20% sparsity in the output layer,
while RMSE drops from 0.632 to 0.629. We would like to include more results and discuss the use of the spike-and-slab
prior in the style of group-Lasso such that a whole pathway will be retained or pruned in the future revision.

# 2. Discussions on larger neural networks.

Extension of the proposed method to larger networks is straightforward. However, as implicitly assumed in our
theory, the convergence of the SGMCMC algorithm is essential. For larger networks, to achieve this convergence,
longer training time might be needed. Existing techniques, such as gradient noise control and temperature tuning, for
accelerating SGMCMC simulations should also be helpful to this proposed method.

| Dataset | Boston | Yacht | Energy | Wine | Concrete |
| Hyperparameters | 100/0.01/1.015 | 10/0.1/1.015 | 10/0.001/1.01 | 10/0.001/1.015 | 10/0.01/1.015 |
| --- | --- | --- | --- | --- | --- |
| SGHMC | 2.840±0.120 | 0.764±0.029 | 1.466±0.058 | 0.654±0.014 | 5.668±0.073 |
| A-SGHMC | 2.887±0.128 | 0.726±0.042 | 1.354±0.044 | 0.632±0.009 | 5.644±0.084 |
| SGHMC-EM | 2.872±0.125 | 0.748±0.048 | 1.412±0.028 | 0.770±0.011 | 5.632±0.057 |
| A-SGHMC-EM | 2.858±0.120 | 0.736±0.036 | 1.402±0.027 | 0.638±0.008 | 5.474±0.096 |
| SGHMC-SA | **2.838±0.115** | **0.746±0.037** | **1.366±0.034** | **0.632±0.010** | **5.372±0.071** |
| A-SGHMC-SA | **2.780±0.108** | **0.716±0.036** | **1.270±0.029** | **0.628±0.008** | 5.438±0.079 |

Table 1: Average testing performance and standard deviation of RMSE (Root Mean Square Error), with $b$ in the Beta distribution, $v_0$ in the SSGL prior, and $r$ in the simulated annealing (Hyperparameters $b/v_0/r$).

[Meta-Review · NeurIPS 2019]

All reviewers acknowledged that the authors addressed their concerns well in the feedback, and agreed to accept the paper.